# OpenReview forum: "mR3: Multilingual Rubric-Agnostic Reward Reasoning Models"
_ICLR.cc/2026/Conference — ICLR 2026 Poster_

### Official Review · Reviewer_hiQ8 · 2025-10-29

**Soundness:** 3
**Presentation:** 3
**Contribution:** 2
**Rating:** 2
**Confidence:** 4

**Summary:**

The paper introduces MR3, a multilingual, rubric-agnostic reward reasoning model trained on 72 languages. It studies multilingual dataset curation, curriculum strategies, and the role of language in rubrics, reasoning, and evaluation.

**Strengths:**

- Impressive language coverage and strong results across diverse multilingual benchmarks.
- Thorough study on how instruction and reasoning language affect performance.
- Clear, well-organized methodology and strong experimental validation.
- Open-sourcing of models and datasets is valuable for the community.

**Weaknesses:**

- Gains are minuscule and not clear if significant, contribution is not novel enough.
- Heavy reliance on GPT-generated rubrics and reasoning may limit true novelty.
- Rubric generation pipeline and translation details could use stronger validation.
- Limited exploration of low-resource languages beyond aggregate results.
- Analysis could go deeper into why certain multilingual strategies work.

**Questions:**

- How well does MR3 generalize to unseen or code-switched languages?
- Could multilingual reasoning data generated by smaller models reduce reliance on large teachers?
- Any evidence that MR3’s reasoning traces improve human interpretability in practice?
- How robust is MR3 to noisy or inconsistent rubrics?

---

> ### Author Response · Authors · 2025-11-27
> **Response to Reviewer hiQ8 (Part 1)**
>
> Thank you for your insightful questions; they were instrumental in helping us clarify and strengthen our paper. We have updated the manuscript based on your feedback. Our responses, along with specific details of the corresponding changes, are provided below.
>
> ## **Response to W1**
>
> We appreciate the reviewer’s time and respectfully disagree with the characterization that our work “Gains are minuscule and not clearly significant; contribution not novel.” Our paper offers (i) a task-agnostic framework for multilingual rubric-based reward reasoning with explicit reasoning traces, (ii) the broadest multilingual training/evaluation setup to date (72 languages), and (iii) a systematic study of data selection and curriculum strategies across languages and supervision choices that go beyond an engineering build of an SFT model.
>
> ### **(1) Gains are not minuscule**
>
> To reiterate, our results provide clear and consistent evidence that the improvements are substantial - surpassing both comparable baselines and even significantly larger models. Here, we provide the mean and standard deviation across 5 different runs. Bolded numbers are the best among a benchmark dataset.
>
> | Model                | m-RewardBench (23 langs) | RewardBench (1 lang) | MM-Eval (18 langs) | IndoPref (1 lang) |  INCLUDE-base-44 Acc. (44 langs) | MGSM Acc. (11 langs) | RTP-LX F1. (27 langs) |
> | -------------------- | ------------------------ | -------------------- | ------------------ | ----------------- |------------------------ | -------------------- | ------------------ |
> | Qwen3-4B             | 84.51 ± 0.08             | 88.04 ± 0.37         | 80.07 ± 0.52       | 68.80 ± 0.32      | 61.54 ± 0.22 | 90.34 ± 0.14 | 84.10 ± 0.16 |
> | Qwen3-8B             | 86.57 ± 0.06             | 88.77 ± 0.13         | 81.95 ± 0.31       | 72.30 ± 0.45      | 66.55 ± 0.09 | 92.52 ± 0.18 | 78.41 ± 1.23 |
> | Qwen3-14B            | 88.46 ± 0.12             | 89.72 ± 0.42         | 84.33 ± 0.35       | 73.22 ± 0.25      | 69.7 ± 0.14 | 93.49 ± 0.16 | 78.36 ± 0.49 |
> | GPT-OSS-20B          | 86.66 ± 0.30             | 87.81 ± 0.39         | 82.03 ± 1.25       | 69.71 ± 0.43      | 62.55 ± 0.28 | 92.57 ± 0.22 | 90.60 ± 0.2 |
> | GPT-OSS-120B         | 89.05 ± 0.06           | 90.30 ± 0.50         | 85.01 ± 0.24     | 72.15 ± 0.15      |  **71.65** ± 0.21 | **94.71** ± 0.17 | 91.32 ± 0.1 |
> | DeepSeek-R1-14B      | 68.53 ± 1.62             | 70.91 ± 1.48         | 58.77 ± 2.10       | 55.19 ± 2.73      | 21.78 ± 1.83 | 69.81 ± 3.15 | 83.83 ± 0.82 |
> | Qwen2.5-14B-Instruct | 77.21 ± 0.06             | 80.64 ± 0.28         | 78.00 ± 0.35       | 70.04 ± 0.26      | 57.93 ± 0.15 | 84.93 ± 0.21 | **91.34** ± 0.14 |
> | mR3-Qwen3-4B             | 87.61 ± 0.17     | 89.74 ± 0.52 | 82.62 ± 0.51     | 72.22 ± 0.25     | 63.01 ± 0.13 | 91.20 ± 0.22 | 88.20 ± 0.07 |
> | mR3-Qwen3-8B             | 88.44 ± 0.07     | 90.50 ± 0.25 | 84.84 ± 0.37     | 72.86 ± 0.16   | 67.72 ± 0.16 | 93.20 ± 0.2 | 90.03 ± 0.12 |
> | mR3-Qwen3-14B            | **89.18 ± 0.08** | **90.79** ± 0.25 | **86.05** ± 0.18 | **74.14** ± 0.26 | 70.61	 ± 0.26	| 94.00  ± 0.07 | 90.19  ± 0.1 |
> | mR3-DeepSeek-R1-14B      | 87.12 ± 0.37     | 88.73 ± 0.95 | 81.85 ± 0.38     | 70.11 ± 0.98     |  60.37 ± 0.25 | 89.48 ± 0.27 | 89.53 ± 0.34 |
> | mR3-Qwen2.5-14B-Instruct | 85.41 ± 0.49     | 88.21 ± 0.51 | 81.51 ± 0.64     | 68.10 ± 0.55     | 63.72 ± 0.51 | 90.41 ± 0.16 | 90.38 ± 0.21 |
>
> **Our models outperform the base models of comparable size by a clear margin, and even surpass substantially larger models such as the 120B GPT-OSS-120B (up to 9x larger).** This demonstrates that our gains are not minuscule; moreover, the low standard deviations indicate that our improvements are statistically meaningful relative to the baselines.

---

> > ### Author Response · Authors · 2025-11-27
> > **Response to Reviewer hiQ8 (Part 2)**
> >
> > ## **Response to W1 (Continuation)**
> >
> > ### **The Novelty of Our Contributions to Multilingual Reward Models is Significant**
> >
> > We wish to re-emphasize the technical and novel contributions of our work:
> > - We present mR3, a task-agnostic framework for training massively multilingual reasoning reward models that leverages fine-grained rubrics, either human-crafted or LLM-generated, for controllable and interpretable scoring. mR3 outperforms existing reward models and achieves performance comparable to much larger models (e.g., mR3-Qwen-14B vs. GPT-OSS-120B) while being up to 9× smaller.
> > - We construct a large, diverse multilingual dataset covering 72 languages from a wide range of sources to train mR3 (Table 1), representing the broadest language coverage to date. We also develop a comprehensive benchmark to evaluate our models across multiple tasks. Upon acceptance, we will open-source the trained models, evaluation code, and datasets.
> > - We systematically investigate dataset selection and curriculum learning strategies along three key dimensions: (i) instruction and rubric language, (ii) response and reasoning language, and (iii) methods for improving target-language reasoning. Our results show that while English remains the most effective prompting language, targeted multilingual training enables mR3 to process target-language inputs more robustly, producing more accurate reasoning and evaluations.
> > - We conduct off-policy preference optimization experiments to showcase our models’ strengths in RL-based optimization. Additionally, we evaluate the quality of our reasoning traces and rubrics through human assessments, where annotators frequently prefer mR3 models over existing reward models, including on extremely low-resource languages that are unseen to the training data.
> >
> > In the following sections, we try to summarize and extend our analyses across multiple orthogonal factors that clarify the sources of improvement and justify our design choices. At the very end of our responses, we include a concise summary of these ablations for your convenience, with full results provided in the Appendix.
> >
> > ### **Decision on Training Dataset Construction**
> >
> > **(1) Data-scale ablation.** We systematically vary the training set size (10K → 25K → 50K → 100K) and observe that larger training sizes consistently improve overall performance. In our experiments, the 100K dataset yields the best results compared to smaller sizes; we attribute this to the large language breadth in our training set (72 languages), which requires substantial data to achieve robust multilingual coverage.
> >
> > **(2) Teacher model quality.** We evaluate the effect of teacher strength on student performance by performing training with a weaker teacher (gpt-oss-20b) and find that **higher-quality teacher reasoning traces result in stronger student reward models, particularly on multilingual benchmarks.**
> >
> > **(3) Curriculum-learning ablation.** We compare six curriculum strategies and select *EasyToHard* because it **performs best on the validation set.** We also verify that this choice generalizes to the test set (which was not used for curriculum selection).
> > We also add an additional note that the previous study (R3 [1]) has already explored the impact of different data sources and dataset components, including rubrics, explanations, and reasoning traces, so our decision regarding dataset construction also builds on that. We also add more older base models to provide additional comparisons with prior works.
> >
> > ### **Decision on Training Strategy**
> >
> > **(4) Comparison between RL and SFT:** We additionally experiment with RLVR (RM-R1–style) using a matched data budget (50K SFT + 50K RLVR). **RLVR does not outperform SFT;** in fact, it degrades multilingual rubric reasoning on our dataset. This is largely because **RLVR provides only correctness-based signals and thus cannot enforce rubric-following behavior,** which was what we discovered upon inspection. We also note that RLVR is **computationally expensive** (2 days on 16×H100 GPUs for Qwen3-4B), whereas 100K SFT requires only 8 hours on 4×H100 GPUs, yet SFT still achieves better results. This analysis explains why SFT is our preferred approach for multilingual, rubric-based reasoning, given our training setup.
> >
> > ### **References**
> >
> > [1] Anugraha, D., Tang, Z., Miranda, L. J. V., Zhao, H., Farhansyah, M. R., Kuwanto, G., ... & Winata, G. I. (2025). R3: Robust rubric-agnostic reward models. arXiv preprint arXiv:2505.13388.

---

> ### Author Response · Authors · 2025-11-27
> **Response to Reviewer hiQ8 (Part 3)**
>
> ## **Response to W1 (Continuation)**
>
> ### **Controlled Study on Multilinguality**
>
> **(5) Supervision-language and multilinguality analyses:**
>
> We conducted controlled multilingual comparisons across four settings for our mR3 models:
> - English instruction + English reasoning
> - Target-language instruction + English reasoning
> - Target-language instruction + target-language reasoning
> - Target-language instruction + target-language reasoning (translated from English reasoning)
>
> As shown in Figure 3, our **mR3 model with English instruction and English reasoning performs best overall, followed by mR3 with target-language instruction and English reasoning, and then mR3 with target-language instruction and target-language reasoning.**
> Importantly, even though target-language reasoning is not the strongest among the mR3 configurations, **it still substantially outperforms the base model, whether the base model reasons in English or directly in the target language.** This highlights that multilingual supervision in mR3 meaningfully improves target-language reasoning quality, demonstrating the robustness and effectiveness of our approach, while providing better interpretability for non-English speakers.
>
> We further compare translated reasoning to language-forced reasoning and show that language forcing yields more faithful reasoning traces.
>
> **(6) Human evaluation of Multilingual Reasoning and Rubrics Quality:**  We conduct a human evaluation across 12 high-, medium-, and low-resource languages using 19 annotators (with additional evaluations ongoing). **Note that all low-resource languages included in the human evaluation (Javanese, Albanian, and Telugu) are entirely unseen in the mR3 training data.** Annotators rate the reasoning of our models, base models, and the training datasets along two dimensions: factual correctness and logical coherence. Overall, annotators report that our models produce more faithful reasoning than the base model by being more succinct, fluent, coherent, and culturally aligned.
>
> The annotators also rate the plausability, scorability, and translation quality of the rubrics and found that the rubrics are of high quality.
>
> ---
>
> We believe that these findings, taken together, are enough contributions to be considered novel in releasing a large multilingual setup and instead provide scientific insights into multilingual, rubric-driven reward modeling. We **provide details and study each factor** in training dataset construction and choice of training strategy, which overall stabilizes and enhances training across a large number of languages. We also **standardize the evaluation format (point-wise, pair-wise, and binary) and provide explicit reasoning outputs for interpretability, all within a unified, rubric-driven framework.** Our rubrics and reasonings are also **validated through human evaluation across 17 different native speakers on 12 languages** on different resource levels, including low-resource languages (more annotations are coming!). Our scope is broader than prior work and is framed as empirical research questions with controlled ablations, rather than as system building.
>
> To reiterate, our paper advances the understanding of how to build multilingual, rubric-driven reward models by studying different factors in dataset construction, training strategy, and the interaction between supervision language and reasoning quality. These decisions allow us to achieve state-of-the-art performance on multilingual reward-model benchmarks while being up to 9× smaller than prior baselines. We will add a "sources of improvement" subsection that cross-references these results and findings, as well as the summary of the supervision-language findings, to make the story more explicit.

---

> ### Author Response · Authors · 2025-11-27
> **Response to Reviewer hiQ8 (Part 4)**
>
> ## **Response to W2 and W3**
>
> We appreciate the concern regarding reliance on synthetic rubrics and reasoning. **While the significant performance gains discussed in Response to W1 demonstrate the empirical effectiveness of our pipeline,** we further validate the intrinsic quality and robustness of these synthetic artifacts through two complementary rigorous approaches: **(1) Human validation of rubric and reasoning quality, and (2) Out-of-Distribution (OOD) testing to verify model robustness.**
>
> ### **(1) Human validation of rubric and reasoning quality:**
>
> In our human studies, we asked 17 native speakers of 12 high-, medium-, and low-resource languages to assess the quality of rubrics used in our framework, training reasoning traces, and evaluation reasoning traces for different 14B models. **Scores are measured between 1-3 (3 means the highest quality).**
>
> **Rubrics are of very high quality**: On average, rubrics score 2.93 ± 0.09 on plausibility, 2.72 ± 0.29 on score-ability, and 2.72 ± 0.35 on translation quality. This indicates that the rubrics we have are of very high quality.
>
> **Training reasoning is of high quality especially in English**: We evaluate logical coherence and factual correctness of reasoning traces in our training data, which is distilled from gpt-oss-120b, and found in general English reasoning (factual: 2.97 ± 0.06; logical: 2.81 ± 0.27) > target reasoning (factual: 2.86 ± 0.27; logical: 2.71 ± 0.42) > translated target reasoning (factual: 2.76 ± 0.3; logical: 2.62 ± 0.30). This is expected as even the teacher models we use are trained mostly in English, and translating English reasoning can result in artifacts of unnaturalness.
>
> **Eval reasoning improves over baseline; Target reasoning often preferred over English reasoning**: We ask annotators to rate evaluation reasoning traces across 12 languages using the same criteria as above. Again, note that all low-resource languages included in the human evaluation (Javanese, Albanian, and Telugu) are entirely unseen in the mR3 training data. We find that our mR3 model with target-language reasoning (factual: 2.78 ± 0.30; logical: 2.67 ± 0.45) outperforms the equivalent Qwen3 baseline (factual: 2.06 ± 0.69; logical: 2.05 ± 0.71), indicating **significant improvement in reasoning faithfulness compared to the base model.** Our annotators report that our models produce more faithful reasoning than the base model by being more succinct, fluent, coherent, and culturally aligned. Several annotators also noted that they preferred mR3 outputs that reasoned in the target language over those that reasoned in English, as target-language reasoning often better captures nuance and cultural context.
>
> Based on the human annotation findings above, our native-speaker annotators consistently rated the generated rubrics, translations, and reasoning traces as high quality. This suggests that the reliance is acceptable based on stronger validations.
>
> ### **(2) Out-of-Distribution (OOD) Testing (Robustness Validation):**
>
> To ensure the model learned robust reasoning rather than rubric-specific shortcuts or GPT artifacts, we tested on distinct OOD scenarios:
> - Domain Shift: We evaluated on domains not seen in training (coding, brainstorming, linguistics) using unseen benchmarks (IndoPref, m-reward-bench, MM-Eval).
> - Rubric Shift: The rubrics in evaluation datasets differ from rubrics used in training (e.g., varying definitions of "safety" between PolyGuard and RTP-LX), ensuring the model cannot "game" the metric.
> - Language Shift: We tested on unseen languages (Telugu, Albanian, Malayalam, and Nepali) on INCLUDE-base-44 and MGSM, while also providing human evaluations on 3 unseen languages (Telugu, Albanian, and Javanese). As detailed in the response to W4 later, mR3 is preferred over the base model even in languages absent from training, on both benchmarks and human evaluations.
>
> These OOD generalizations further support the robustness of our data pipeline and the resulting models.

---

> > ### Author Response · Authors · 2025-11-27
> > **Response to Reviewer hiQ8 (Part 5)**
> >
> > ## **Response to W4**
> >
> > Thank you for the reviewer’s comment. To address this comment, we provide (1) qualitative human validation of mR3 reasoning quality in these languages, and (2) a quantitative breakdown of performance in low-resource languages, including for unseen/OOD languages, on benchmarks with parallel data.
> >
> > ### **(1) Human Validation on Low-Resource Languages:**
> >
> > As described in our response to W2 and W3, we perform a human validation by having 17 native speakers across 12 languages. Here, we have 3 low-resource languages, which are Albanian, Javanese, and Telugu, **all are OOD/unseen to the training set.** Quantitatively, mR3 with target-language reasoning (factual: 2.48 ± 0.56; logical: 2.40 ± 0.61) substantially outperforms the Qwen3 baseline (factual: 1.63 ± 0.87; logical: 1.76 ± 0.90). **Qualitatively, annotators report that mR3 produces more faithful reasoning by being more succinct, fluent, coherent, and culturally aligned.** We also highlight an example of comparison between mR3 reasonings and Qwen3 reasonings on Telugu, which is OOD/unseen during training. Annotators note that the Qwen3 base model often fails to produce substantive Telugu reasoning, frequently restating its intent to reason rather than providing actual logical steps. In contrast, mR3 generates generally coherent Telugu reasoning; even when its Telugu-to-English translation becomes literal, the reasoning remains comprehensible. (See Table 33 for the illustrative example.) These examples and analyses are included in Appendix J.4.
> >
> > ### **(2) Quantitative Breakdown Through Benchmarks:**
> >
> > We report performance on INCLUDE-base-44, MGSM, and RTP-LX restricted to low-resource languages. We also report another performance on INCLUDE-base-44 and MGSM, which are strictly for unseen languages. To directly evaluate target-language reasoning, we use target-language prompts and target-language reasoning traces. As shown below, **all mR3 variants consistently and substantially outperform their corresponding base models across all benchmarks.** Moreover, the mR3 models often obtain the best overall performance, reinforcing that target-language reasoning improves accuracy and better aligns with native speaker expectations.
> >
> > | Model | INCLUDE-base-44 (LRL only) | MGSM (LRL only) | RTP-LX (LRL only) |
> > |-------|---------------|------------|--------------|
> > | Qwen3-4B | 42.64 ± 0.25 | 35.27 ± 1.17 | 45.22 ± 1.54 |
> > | **mR3-Qwen3-4B** | 56.30 ± 1.13 | 74.60 ± 0.80 | 77.52 ± 8.89 |
> > | Qwen3-8B | 45.96 ± 0.83 | 50.76 ± 6.96 | 37.87 ± 3.59 |
> > | **mR3-Qwen3-8B** | 61.20 ± 0.54 | 80.46 ± 1.61 | 83.02 ± 2.75 |
> > | Qwen3-14B | 54.68 ± 1.67 | 63.18 ± 4.63 | 53.82 ± 4.04 |
> > | **mR3-Qwen3-14B** | **65.00** ± 1.13 | **85.60** ± 0.82 | **78.12** ± 1.88 |
> > | GPT-OSS-20B | 51.49 ± 1.14 | 67.64 ± 1.14 | 71.27 ± 4.21 |
> >
> > | Model             | INCLUDE-base-44 (LRL and Unseen only) | MGSM (LRL and Unseen only) |
> > | ----------------- | ------------------------------------- | -------------------------- |
> > | Qwen3-4B          | 38.81 ± 0.55                          | 61.75 ± 0.55               |
> > | **mR3-Qwen3-4B**  | 59.05 ± 1.15                          | 82.44 ± 1.08               |
> > | Qwen3-8B          | 41.83 ± 1.08                          | 74.88 ± 1.07               |
> > | **mR3-Qwen3-8B**  | 59.54 ± 1.66                          | 86.64 ± 0.46               |
> > | Qwen3-14B         | 49.40 ± 2.50                          | 80.50 ± 2.22               |
> > | **mR3-Qwen3-14B** | **64.35** ± 1.77                          | **88.68** ± 1.09               |
> > | GPT-OSS-20B       | 50.00 ± 1.61                          | 78.40 ± 0.79               |

---

> > > ### Author Response · Authors · 2025-11-27
> > > **Response to Reviewer hiQ8 (Part 6)**
> > >
> > > ## **Response to W5**
> > >
> > > **In our response to W1, we have added a lot of analysis regarding the sources of improvement and justify our design choices. Please mainly refer to our response to W1.** To summarize, our multilingual strategies help the mR3 models to follow multilingual instructions and reason in target languages based on the high-quality SFT data (validated by humans) through an extensive process of training data collection.
> > >
> > > ## Response to Q1
> > >
> > > **We have discussed this in our response to W4, please refer to that section.**
> > >
> > > ## Response to Q2
> > >
> > > **We have partially discussed this in our response to W1, but here is the following detail:**
> > >
> > > We evaluated the impact of using a weaker teacher model, gpt-oss-20B, compared to our normal teacher, gpt-oss-120b. The results are averaged over 5 seeds.
> > >
> > > | Setting                     | m-Reward Bench | Reward Bench | IndoPref     | MM Eval      | INCLUDE      | mgsm         | RTP-LX       | Overall      |
> > > | --------------------------- | -------------- | ------------ | ------------ | ------------ | ------------ | ------------ | ------------ | ------------ |
> > > | Qwen3-4B (Normal Teacher)  | 87.61 ± 0.17 | 89.74 ± 0.52 | 72.22 ± 0.25 | 82.62 ± 0.51 | 63.01 ± 0.13 | 91.20 ± 0.22 | 88.20 ± 0.07 | 82.09 ± 0.27 |
> > > | Qwen3-4B (Weak Teacher)     | 84.63 ± 0.29   | 87.44 ± 0.30 | 69.95 ± 0.23 | 78.54 ± 0.50 | 54.24 ± 0.10 | 84.59 ± 0.18 | 88.06 ± 0.03 | 78.21 ± 0.23 |
> > > | Qwen3-4B (Baseline) | 84.51 ± 0.08   | 88.04 ± 0.37 | 68.80 ± 0.32 | 80.07 ± 0.52 | 61.54 ± 0.22 | 90.34 ± 0.14 | 84.10 ± 0.16 | 79.63 ± 0.26 |
> > >
> > > We note that teacher quality plays a crucial role in SFT since the model learns directly from the outputs of the teacher, so the quality and consistency of these outputs somewhat define the upper bound of the student model’s performance. A weaker teacher may still provide useful signals, but its reasoning, factual accuracy, and consistency are inherently lower, especially in this case when the performance of the teacher model is comparable with the student model. This can lead to less stable or suboptimal learning outcomes. On the other hand, **a high-quality teacher provides strong, coherent, and accurate reasoning traces, which allow the student to better capture complex reasoning patterns across tasks and languages.**
> > >
> > > ## Response to Q3
> > >
> > > **We have partially discussed this in our response to W1, W2, W3, and W4.** To summarize, our annotators report that our models produce more faithful reasoning than the base model by being more succinct, fluent, coherent, and culturally aligned. Several annotators also noted that they preferred mR3 outputs that reasoned in the target language over those that reasoned in English, as target-language reasoning often better captures nuance and cultural context. Therefore, we believe that mR3 reasoning traces can potentially improve human interpretability based on some annotators' feedback.
> > >
> > > ## Response to Q4
> > >
> > > **We have partially addressed this in W2 and W3.** Recall that the evaluation rubrics differ from those used during training. For example, the definition of “safety” in PolyGuard is not the same as in RTP-LX. This prevents the model from overfitting to any specific rubric format. Moreover, each evaluation rubric is paraphrased into three variants and randomly selected across runs, introducing controlled “noise”. The resulting standard deviations therefore implicitly capture robustness to rubric variation. As shown in our results, the mR3 models maintain strong and stable performance despite this noise, indicating robustness to rubric differences.
> > >
> > > In terms of inconsistent rubrics, mR3 is designed to follow instructions faithfully. If the rubrics are inconsistent, the model's performance will naturally degrade, mirroring the behavior we would have seen when human evaluators are faced with similar ambiguity.
> > >
> > > We believe that robustness to inconsistent rubrics is fundamentally a prompt engineering challenge, not a modeling flaw. Attempting to make the model 'guess' the correct intent behind a noisy rubric risks introducing unverified bias. Therefore, we treat rubric quality as an external dependency: if the rubric is poor, the evaluation system should ideally flag the ambiguity or fail, rather than attempting to 'silently fix' it. Resolving rubric inconsistency is beyond the scope of this study, as it requires distinct methodologies, such as automated prompt optimization or interactive Human-AI clarification loops during rubric design, rather than changes to the reward model architecture itself.
> > >
> > > Thank you again for your valuable feedback. We hope these clarifications effectively address your concerns. If our responses have satisfactorily addressed your concerns, we kindly ask you to reconsider your evaluation accordingly.

---

> ### Author Response · Authors · 2025-11-27
> **Quick Reference for Experiments for Reviewer hiQ8 (Part 1)**
>
> **As part of our reference, we include a concise summary of these ablations for your convenience, with full results provided in the Appendix.**
>
> ### **Ablation 1: Data Sizes**
>
> We evaluate the effect of dataset size using subsets of 10K, 25K, 50K, and 100K samples and use these as the training datasets for Qwen3-4B. The standard deviations are obtained from the 5 runs to provide more insights into the results.
>
> | Setting           | m-RewardBench  | RewardBench  | IndoPref  | MM-Eval  | INCLUDE-base-44  | MGSM  | RTP-LX  | Overall |
> |------------------|------------------|----------------|--------------|-------------|------------|----------|-----------|-------------|
> | Qwen3-4B (10K)   | 86.05 ± 0.03     | 88.48 ± 0.39   | 69.98 ± 0.36 | 79.26 ± 0.09 | 60.17 ± 0.17 | 89.79 ± 0.20 | **89.67** ± 0.10 | 80.49 ± 0.19 |
> | Qwen3-4B (25K)   | 85.99 ± 0.16     | 88.96 ± 0.70   | 70.14 ± 0.71 | 79.62 ± 0.56 | 60.06 ± 0.38 | 88.12 ± 0.16 | 89.64 ± 0.23 | 80.36 ± 0.41 |
> | Qwen3-4B (50K)   | 86.43 ± 0.08     | 88.79 ± 0.45   | 70.90 ± 0.36 | 80.22 ± 0.38 | 60.29 ± 0.11 | 88.23 ± 0.21 | 89.03 ± 0.11 | 80.56 ± 0.24 |
> | Qwen3-4B (100K)  | **87.61** ± 0.17     | **89.74** ± 0.52   | **72.22** ± 0.25 | **82.62** ± 0.51 | **63.01** ± 0.13 | **91.20** ± 0.22 | 88.20 ± 0.07 | **82.09** ± 0.27 |
>
> We observe a significant improvement when scaling the dataset to 100K, likely because our training data spans 72 languages, so smaller dataset sizes are less effective for robust multilingual coverage. While adding even more data may further improve performance, we are constrained by a budget of 100K.
>
> ### **Ablation 2: Curriculum Learning**
>
> As explained in the paper, we studied six curriculum ordering strategies:
> - Random Shuffle
> - EasyToHard
> - HardToEasy
> - English-First
> - English-First + EasyToHard
> - English-First + HardToEasy
>
> We selected EasyToHard as our final curriculum based on validation-set performance (HelpSteer3), where it consistently outperformed alternative curricula across 5 seeds.
>
> | Model                          | Strategy               | Kendall Tau |
> |--------------------------------|----------------------|-------------|
> | Qwen3-4B | EasyToHard (Our Final Curriculum)        | **0.4779** ± 0.0114      |
> | Qwen3-4B | Random               | 0.4583  ± 0.0083      |
> | Qwen3-4B | HardToEasy           | 0.4647 ± 0.0092      |
> | Qwen3-4B | English-First             | 0.4516 ± 0.0049      |
> | Qwen3-4B | English-First + EasyToHard  | 0.3800 ± 0.0086      |
> | Qwen3-4B | English-First + HardToEasy  | 0.4629 ± 0.0036      |
>
> For additional insight, we evaluated these curricula on the test set (without using it to select the curriculum). EasyToHard generally outperforms other strategies across most metrics, with minor exceptions in some cases. We also provide standard deviations over 5 seeds to confirm that the differences are meaningful.
>
> | Setting                             | m-RewardBench | RewardBench | IndoPref | MM-Eval | INCLUDE-base-44 | MGSM | RTP-LX | Overall |
> |-------------------------------------|------------------|----------------|--------------|-------------|------------|----------|-----------|-------------|
> | Qwen3-4B EasyToHard (Our Final Curriculum) | **87.61** ± 0.17 | 89.74 ± 0.52 | **72.22** ± 0.25 | **82.62** ± 0.51 | **63.01** ± 0.13 | **91.20** ± 0.22 | 88.20 ± 0.07 | **82.09** ± 0.27 |
> | Qwen3-4B Random Shuffle             | 87.09 ± 0.17 | 89.56 ± 0.40 | 71.44 ± 0.57 | 81.95 ± 0.31 | 62.19 ± 0.10 | 90.44 ± 0.17 | 88.97 ± 0.13 | 81.66 ± 0.26 |
> | Qwen3-4B HardToEasy                 | 87.15 ± 0.02 | 89.73 ± 0.44 | 71.30 ± 0.17 | 81.68 ± 0.37 | 61.59 ± 1.05 | 90.60 ± 0.45 | **88.98** ± 0.18 | 81.58 ± 0.38 |
> | Qwen3-4B English-First              | 87.03 ± 0.12 | **89.84** ± 0.25 | 71.50 ± 0.56 | 82.19 ± 0.22 | 62.11 ± 0.09 | 90.56 ± 0.23 | 88.72 ± 0.13 | 81.71 ± 0.23 |
> | Qwen3-4B English-First + EasyToHard   | 86.20 ± 0.07 | 88.30 ± 0.43 | 71.41 ± 0.03 | 79.84 ± 0.46 | 59.60 ± 0.16 | 87.96 ± 0.06 | 88.43 ± 0.04 | 80.25 ± 0.18 |
> | Qwen3-4B English-First + HardToEasy   | 86.98 ± 0.12 | 89.11 ± 0.35 | 71.89 ± 0.51 | 81.66 ± 0.69 | 62.33 ± 0.21 | 90.44 ± 0.15 | 88.65 ± 0.16 | 81.58 ± 0.31 |

---

> > ### Author Response · Authors · 2025-11-27
> > **Quick Reference for Experiments for Reviewer hiQ8 (Part 2)**
> >
> > ### **Ablation 3: Weaker Base Models**
> >
> > We evaluated models using Qwen2.5-14B-Instruct and DeepSeek-R1-14B as base models, which provide a fairer comparison with prior works’ base models, such as m-prometheus-14B and DeepSeek-R1-14B. Standard deviations are reported over 5 seeds where possible.
> >
> > | Setting | m-RewardBench | RewardBench | IndoPref | MM-Eval |
> > |---------|------------------|----------------|--------------|-------------|
> > | Qwen2.5-14B-Instruct | 77.21 ± 0.06 | 80.64 ± 0.28 | 70.04 ± 0.26 | 78.00 ± 0.35 |
> > | DeepSeek-R1-14B | 68.53 ± 1.62 | 70.91 ± 1.48 | 55.19 ± 2.73 | 58.77 ± 2.10 |
> > | RM-R1 14B | 85.49 ± 0.55 | 88.51 | 66.42 ± 0.12 | 74.12 ± 1.27 |
> > | prometheus-7b-v2.0 | 67.31 | 72.05 | 57.41 ± 0.72 | 60.90 |
> > | prometheus-8x7b-v2.0 | 75.15 | 74.06 | 58.38 ± 0.70 | 64.34 |
> > | m-prometheus-7b-v2.0 | 77.54 | 76.84 | 60.08 ± 0.66 | 69.66 |
> > | m-prometheus-14b-v2.0 | 79.51 | 79.67 | 48.16 ± 0.11 | 77.26 |
> > | mR3-Qwen2.5-14B-Instruct | 85.41 ± 0.49 | 88.21 ± 0.51 | 68.10 ± 0.55 | 81.51 ± 0.64 |
> > | mR3-DeepSeek-R1-14B | **87.12** ± 0.37 | **88.73** ± 0.95 | **70.11** ± 0.98 | **81.85** ± 0.38 |
> >
> > We can observe that mR3 models consistently outperform baseline models, especially on multilingual benchmarks (i.e., m-RewardBench, IndoPref, and MM-Eval). For English-only benchmarks (RewardBench), mR3 is still better than RM-R1 14B and m-prometheus 14B. Note that some standard deviations are not reported because they were not included in the corresponding works.
> >
> > ### **Ablation 4: Weaker Teacher Models**
> >
> > We also evaluated the impact of using a weaker teacher model, gpt-oss-20B, compared to our normal teacher, gpt-oss-120b. Same with all other results, these runs are averaged over 5 seeds.
> >
> > | Setting                     | m-RewardBench | RewardBench | IndoPref     | MM-Eval      | INCLUDE-base-44     | MGSM        | RTP-LX       | Overall      |
> > | --------------------------- | -------------- | ------------ | ------------ | ------------ | ------------ | ------------ | ------------ | ------------ |
> > | Qwen3-4B (Our Normal Teacher)  | **87.61** ± 0.17 | **89.74** ± 0.52 | **72.22** ± 0.25 | **82.62** ± 0.51 | **63.01** ± 0.13 | **91.20** ± 0.22 | **88.20** ± 0.07 | **82.09** ± 0.27 |
> > | Qwen3-4B (Weak Teacher)     | 84.63 ± 0.29   | 87.44 ± 0.30 | 69.95 ± 0.23 | 78.54 ± 0.50 | 54.24 ± 0.10 | 84.59 ± 0.18 | 88.06 ± 0.03 | 78.21 ± 0.23 |
> > | Qwen3-4B (Baseline) | 84.51 ± 0.08   | 88.04 ± 0.37 | 68.80 ± 0.32 | 80.07 ± 0.52 | 61.54 ± 0.22 | 90.34 ± 0.14 | 84.10 ± 0.16 | 79.63 ± 0.26 |
> >
> > We note that teacher quality plays a crucial role in SFT since the model learns directly from the outputs of the teacher, so the quality and consistency of these outputs somewhat define the upper bound of the student model’s performance. A weaker teacher may still provide useful signals, but its reasoning, factual accuracy, and consistency are inherently lower, especially in this case when the performance of the teacher model is comparable with the student model. This can lead to less stable or suboptimal learning outcomes. On the other hand, a **high-quality teacher provides strong, coherent, and accurate reasoning traces, especially when we want to encourage the model to reason through rubrics, which allow the student to better capture complex reasoning patterns across tasks and languages.**

---

> > > ### Author Response · Authors · 2025-11-27
> > > **Quick Reference for Experiments for Reviewer hiQ8 (Part 3)**
> > >
> > > ### Additional Study between RL and SFT
> > >
> > > Based on the suggestion of Reviewer KCai, we try to compare SFT with an RL paradigm. We conducted experiments using RLVR with GRPO, following the approach of RM-R1 and similar to Nemotron, where the model receives a reward of +1 for a correct answer and -1 for an incorrect answer. We used Qwen3-4B as the base model, starting from the 50K checkpoint from our data scaling ablation and applying RLVR to the remaining 50K examples, to maintain a total of 100K training samples, comparable to full SFT training. **After extensive hyperparameter tuning over the past few weeks, the best results we obtained with RLVR still fall short of the SFT baseline, except on RTP-LX.** Here we show the result of our best possible checkpoint over 5 seeds:
> > >
> > > | Setting                       | m-RewardBench | RewardBench | IndoPref     | MM-Eval      | INCLUDE-base-44      | MGSM         | RTP-LX       | Overall      |
> > > | ----------------------------- | -------------- | ------------ | ------------ | ------------ | ------------ | ------------ | ------------ | ------------ |
> > > | Qwen3-4B (50K SFT)            | 86.43 ± 0.08   | 88.79 ± 0.45 | 70.90 ± 0.36 | 80.22 ± 0.38 | 60.29 ± 0.11 | 88.23 ± 0.21 | 89.03 ± 0.11 | 80.56 ± 0.24 |
> > > | Qwen3-4B (100K SFT)           | **87.61** ± 0.17   | **89.74** ± 0.52 | **72.22** ± 0.25 | **82.62** ± 0.51 | **63.01** ± 0.13 | **91.20** ± 0.22 | 88.20 ± 0.07 | **82.09** ± 0.27 |
> > > | Qwen3-4B (50K SFT + 50K RLVR) | 84.91 ± 0.05   | 87.72 ± 0.56 | 66.12 ± 0.50 | 80.57 ± 0.52 | 61.42 ± 0.21 | 90.17 ± 0.16 | **92.10** ± 0.08 | 80.43 ± 0.30 |
> > >
> > > We also performed qualitative analysis of the reasoning behavior. We found that RLVR does not effectively utilize rubrics during reasoning. This is somewhat expected because RLVR provides reward feedback only based on the correctness of the final answer, without evaluating the quality of the reasoning process. Consequently, it is **plausible that the model could arrive at correct answers without following the rubric properly, as it is not necessary in order to obtain a good reward.**
> > >
> > > In contrast, **SFT with high-quality instruction and reasoning data explicitly teaches the model to follow rubrics during reasoning.** This effect is particularly pronounced in multilingual settings, where our human study evaluations show that the SFT-trained models improve their reasoning capabilities across different rubrics and languages. In short, SFT encourages the model to learn structured reasoning and rubric adherence, whereas RLVR primarily optimizes for answer correctness and does not explicitly supervise the model to learn to reason through the provided rubrics.
> > >
> > > Efficiency-wise, **RLVR is significantly more expensive:** training 3 epochs of RLVR on Qwen3-4B with 16 H100 GPUs takes ~2 days, while training 3 epochs of SFT on 100K samples only takes ~8 hours on 4 H100 GPUs. Thus, SFT is not only more effective but also computationally cheaper.
> > >
> > > Finally, prior RL approaches were mostly evaluated in pairwise or limited settings, and it remains unclear how well they generalize to a diverse selection of datasets and tasks. In contrast, our SFT approach demonstrates consistent improvements across multiple multilingual and multi-task benchmarks.

---

> > > > ### Author Response · Authors · 2025-11-27
> > > > **Quick Reference for Experiments for Reviewer hiQ8 (Part 4)**
> > > >
> > > > ### Additional Human Evaluation Study
> > > >
> > > > In our human studies, we asked 17 native speakers of 12 high-, medium-, and low-resource languages to assess the quality of rubrics used in our framework, training reasoning traces, and evaluation reasoning traces for different 14B models. All low-resource languages included in the human evaluation (Javanese, Albanian, and Telugu) are entirely unseen in the mR3 training data. **Scores are measured between 1-3 (3 means the highest quality).**
> > > >
> > > > **Rubrics are of very high quality**: On average, rubrics score 2.93 ± 0.09 on plausibility, 2.72 ± 0.29 on score-ability, and 2.72 ± 0.35 on translation quality. This indicates that the rubrics we have are of very high quality.
> > > >
> > > > **Training reasoning is of high quality especially in English**: We evaluate logical coherence and factual correctness of reasoning traces in our training data, which is distilled from gpt-oss-120b, and found in general English reasoning (factual: 2.97 ± 0.06; logical: 2.81 ± 0.27) > target reasoning (factual: 2.86 ± 0.27 ;logical: 2.71 ± 0.42) > translated target reasoning (factual: 2.76 ± 0.3; logical: 2.62 ± 0.30). This is expected as even the teacher models we use are trained mostly in English, and translating English reasoning can result in artifacts of unnaturalness.
> > > >
> > > > **Eval reasoning improves over baseline; Target reasoning often preferred over English reasoning**: We ask annotators to rate evaluation reasoning traces across 12 languages using the same criteria as above. We find that our mR3 model with target-language reasoning (factual: 2.78 ± 0.30; logical: 2.67 ± 0.45) outperforms the equivalent Qwen3 baseline (factual: 2.06 ± 0.69; logical: 2.05 ± 0.71), indicating **significant improvement in reasoning faithfulness compared to the base model.**
> > > >
> > > > We provide more detail on the study design, metrics, tabled results, and more qualitative analysis including some examples in the Appendix section.

---

### Official Review · Reviewer_oUA2 · 2025-10-29

**Soundness:** 3
**Presentation:** 3
**Contribution:** 3
**Rating:** 4
**Confidence:** 4

**Summary:**

This paper introduces mR3, a family of massively multilingual reward models designed to address the performance gap in using large language models as automated evaluators for non-English text. To train mR3, the authors collected a diverse dataset covering 72 languages. The authors demonstrate that mR3 achieves state-of-the-art performance in multilingual reward modeling benchmarks, with significantly fewer parameters.

**Strengths:**

1. The research topic is interesting and relevant.
2. The authors attached their code in the supplementary material and claim to release the dataset and models upon publication, which promotes reproducibility. The dataset could be a significant contribution to the research community.

**Weaknesses:**

1. The paper lacks scientific findings and insights. It seems more like an engineering effort to build a multilingual reward model using supervised fine-tuning, rather than a research paper that advances understanding in the field. For instance, the authors did not provide any evidence to support their decision regarding curriculum learning. Another example is that the authors did not explain where the improvements come from.
2. The paper is not well-written and hard to follow. The authors conduct a plethora of experiments, which is appreciated, but fail to provide clear organization and explanations, making it difficult to grasp the key contributions and results. Lots of important details are buried in the appendix.
3. The evaluation methodology is potentially flawed. While it is understandable that most existing benchmarks are translated from English, which is beyond the authors' control, it raises the question of whether the improvements on these reward modeling benchmarks truly reflect human preferences in non-English languages. To strengthen the evaluation, the authors could consider conducting human evaluations in multiple languages to validate the model's performance. Additionally, the inclusion of RewardBench and IndoPerf in the evaluation is somewhat problematic, as these benchmarks are limited to English and Indonesian, respectively. If we treat all languages equally, including these benchmarks in the evaluation of a multilingual model is likely to bias the overall results towards English and Indonesian, making the model's performance appear better than it actually is in other languages.
4. mR3 is a family of reward models, but the authors did not use them as the reward model to train any multilingual LLMs. It would be interesting to see how effective mR3 is in training multilingual LLMs compared to other reward models.

**Questions:**

See Weaknesses.

---

> ### Author Response · Authors · 2025-11-27
> **Response to Reviewer oUA2 (Part 1)**
>
> Thank you for the reviewer's response. We appreciate the reviewer's valuable feedback and suggestions for improving our paper. We have revised the paper accordingly and outlined the specific changes below.
>
> ## **Response to W1**
>
> We appreciate the reviewer’s time and respectfully disagree with the characterization that our work “lacks scientific findings and insights”. Our paper offers (i) a task-agnostic framework for multilingual rubric-based reward reasoning with explicit reasoning traces, (ii) the broadest multilingual training/evaluation setup to date (72 languages), and (iii) a systematic study of data selection and curriculum strategies across languages and supervision choices that go beyond an engineering build of an SFT model.
> To address the reviewer’s comment about “no explanation of where improvements come from,” we summarize and extend our analyses across multiple orthogonal factors that clarify the sources of improvement and justify our design choices. **In a separate thread of our responses, we include a concise summary of these ablation results for your convenience, with full results provided in the updated Appendix.**
>
> ### **Decision on Training Dataset Construction**
>
> **(1) Data-scale ablation.** We systematically vary the training set size (10K → 25K → 50K → 100K) and observe that larger training sizes consistently improve overall performance. In our experiments, the 100K dataset yields the best results compared to smaller sizes; we attribute this to the large language breadth in our training set (72 languages), which requires substantial data to achieve robust multilingual coverage.
>
> **(2) Teacher model quality.** We evaluate the effect of teacher strength on student performance by performing training with a weaker teacher (gpt-oss-20b) and find that **higher-quality teacher reasoning traces result in stronger student reward models, particularly on multilingual benchmarks.**
>
> **(3) Curriculum-learning ablation.** We compare six curriculum strategies and select *EasyToHard* because it **performs best on the validation set.** We also verify that this choice generalizes to the test set (which was not used for curriculum selection).
> We also add an additional note that the previous study (R3 [1]) has already explored the impact of different data sources and dataset components, including rubrics, explanations, and reasoning traces, so our decision regarding dataset construction also builds on that. We also add more older base models to provide additional comparisons with prior works.
>
> ### **Decision on Training Strategy**
>
> **(4) Comparison between RL and SFT:** We additionally experiment with RLVR (RM-R1–style) using a matched data budget (50K SFT + 50K RLVR). **RLVR does not outperform SFT;** in fact, it degrades multilingual rubric reasoning on our dataset. This is largely because **RLVR provides only correctness-based signals and thus cannot enforce rubric-following behavior,** which was what we discovered upon inspection. We also note that RLVR is **computationally expensive** (2 days on 16×H100 GPUs for Qwen3-4B), whereas 100K SFT requires only 8 hours on 4×H100 GPUs, yet SFT still achieves better results. This analysis explains why SFT is our preferred approach for multilingual, rubric-based reasoning, given our training setup.
>
> ### **Controlled Study on Multilinguality**
>
> **(5) Supervision-language and multilinguality analyses:**
>
> We conducted controlled multilingual comparisons across four settings for our mR3 models:
> - English instruction + English reasoning
> - Target-language instruction + English reasoning
> - Target-language instruction + target-language reasoning
> - Target-language instruction + target-language reasoning (translated from English reasoning)
>
> As shown in Figure 3, our **mR3 model with English instruction and English reasoning performs best overall, followed by mR3 with target-language instruction and English reasoning, and then mR3 with target-language instruction and target-language reasoning.**
> Importantly, even though target-language reasoning is not the strongest among the mR3 configurations, **it still substantially outperforms the base model, whether the base model reasons in English or directly in the target language.** This highlights that multilingual supervision in mR3 meaningfully improves target-language reasoning quality, demonstrating the robustness and effectiveness of our approach, while providing better interpretability for non-English speakers.
>
> We further compare translated reasoning to language-forced reasoning and show that language forcing yields more faithful reasoning traces.
>
> ### **References**
>
> [1] Anugraha, D., Tang, Z., Miranda, L. J. V., Zhao, H., Farhansyah, M. R., Kuwanto, G., ... & Winata, G. I. (2025). R3: Robust rubric-agnostic reward models. arXiv preprint arXiv:2505.13388.

---

> ### Author Response · Authors · 2025-11-27
> **Response to Reviewer oUA2 (Part 2)**
>
> ### **Response to W1 (Continuation)**
>
> **(6) Human evaluation of Multilingual Reasoning and Rubrics Quality:**  We conduct a human evaluation across 12 high-, medium-, and low-resource languages using 19 annotators (with additional evaluations ongoing). **Note that all low-resource languages included in the human evaluation (Javanese, Albanian, and Telugu) are entirely unseen in the mR3 training data.** Annotators rate the reasoning of our models, base models, and the training datasets along two dimensions: factual correctness and logical coherence. Overall, annotators report that our models produce more faithful reasoning than the base model by being more succinct, fluent, coherent, and culturally aligned.
>
> The annotators also rate the plausability, scorability, and translation quality of the rubrics and found that the rubrics are of high quality.
>
> ---
>
> These findings, taken together, go beyond “just engineering effort” in releasing a large multilingual setup and instead provide scientific insights into multilingual, rubric-driven reward modeling. We **provide details and study each factor** in training dataset construction and choice of training strategy, which overall stabilizes and enhances training across a large number of languages. We also **standardize the evaluation format (point-wise, pair-wise, and binary) and provide explicit reasoning outputs for interpretability, all within a unified, rubric-driven framework.** Our rubrics and reasonings are also **validated through human evaluation across 17 different native speakers on 12 languages** on different resource levels, including low-resource languages (more annotations are coming!). Our scope is broader than prior work and is framed as empirical research questions with controlled ablations, rather than as system building.
>
> To reiterate, our paper advances the understanding of how to build multilingual, rubric-driven reward models by studying different factors in dataset construction, training strategy, and the interaction between supervision language and reasoning quality. These decisions allow us to achieve state-of-the-art performance on multilingual reward-model benchmarks while being up to 9× smaller than prior baselines. We will add a "sources of improvement" subsection that cross-references these results and findings, as well as the summary of the supervision-language findings, to make the story more explicit.
>
> ## **Response to W2**
>
> We appreciate the reviewer’s feedback. We will update the paper to better highlight the key contributions and experimental findings, move essential details from the appendix to the main text, and provide clearer explanations to improve readability. We have also provided **additional details regarding our studies in W1, which document the decision-making behind our dataset construction, training strategies, and evaluations.**
>
> We have written the contributions of our paper in the introduction part of the updated paper as follows:
> - We present mR3, a task-agnostic framework for training massively multilingual reasoning reward models that leverages fine-grained rubrics, either human-crafted or LLM-generated, for controllable and interpretable scoring. mR3 outperforms existing reward models and achieves performance comparable to much larger models (e.g., mR3-Qwen-14B vs. GPT-OSS-120B) while being up to 9× smaller.
> - We construct a large, diverse multilingual dataset covering 72 languages from a wide range of sources to train mR3 (Table 1), representing the broadest language coverage to date. We also develop a comprehensive benchmark to evaluate our models across multiple tasks. Upon acceptance, we will open-source the trained models, evaluation code, and datasets.
> - We systematically investigate dataset selection and curriculum learning strategies along three key dimensions: (i) instruction and rubric language, (ii) response and reasoning language, and (iii) methods for improving target-language reasoning. Our results show that while English remains the most effective prompting language, targeted multilingual training enables mR3 to process target-language inputs more robustly, producing more accurate reasoning and evaluations.
> - We conduct off-policy preference optimization experiments to showcase our models’ strengths in RL-based optimization. Additionally, we evaluate the quality of our reasoning traces and rubrics through human assessments, where annotators frequently prefer mR3 models over existing reward models, including on extremely low-resource languages that are unseen to the training data.
>
> We hope that the revised structure emphasizes the main contributions and experimental insights, making the paper easier to follow while preserving the depth of our empirical studies.

---

> ### Author Response · Authors · 2025-11-27
> **Response to Reviewer oUA2 (Part 3)**
>
> ## **Response to W3**
>
> We thank the reviewer for their concern about the evaluation methodology. We would like to clarify that our evaluation methodology is designed to provide a fair and comprehensive assessment of multilingual reward modeling. To reiterate, our evaluation set includes preference-based benchmarks such as RewardBench (English), m-RewardBench (parallel on 23 languages), MM-Eval (18 languages), and IndoPref (Indonesian), covering 30 unique languages across diverse domains and cultures; MGSM, a multilingual mathematics benchmark in 11 languages; INCLUDE-base-44, a multilingual cultural knowledge benchmark spanning 44 languages; and RTP-LX, a multilingual safety dataset covering 28 languages.
>
> Importantly, all major multilingual datasets (MGSM, INCLUDE-base-44, RTP-LX) are human-annotated and parallel across languages. Therefore, these benchmarks provide a fair and meaningful measure of progress in developing multilingual models across all languages, while accurately reflecting improvements in model capabilities. While some datasets are translated from English (e.g., m-RewardBench), they remain parallel and human-validated, which ensures comparability across languages. Therefore, we believe that by comparing the average within these datasets, we represent each language equally. However, if more details are needed, we provide more detailed results in the Appendix, showing the breakdown of each language, including breakdown per language resource level, showing the level of improvements between high-, medium-, and low-resource languages.
>
> RewardBench and IndoPref are included for additional context as RewardBench provides a reference point in English, while IndoPref offers a high-quality, human-annotated dataset in 10 Indonesian domains, which is valuable for analyzing specific multilingual capabilities and identifying areas of improvement. In addition, we report results for each dataset individually rather than averaging across datasets, except when providing concise summary statistics. All detailed results are available for inspection, and our models consistently perform well on each individual dataset.
>
> Finally, to further strengthen our results, as mentioned in **W1** as well, we conduct a human evaluation across 12 high-, medium-, and low-resource languages using 19 annotators (with additional evaluations ongoing). Annotators rate the reasoning of our models, base models, and the training datasets along two dimensions: factual correctness and logical coherence. Overall, annotators report that our models produce more faithful reasoning than the base model by being more succinct, fluent, coherent, and culturally aligned. As a reminder, a more detailed summary can be found in a separate thread of our responses, while the full results and analysis can be found in the Appendix section of the updated paper.

---

> ### Author Response · Authors · 2025-11-27
> **Response to Reviewer oUA2 (Part 4)**
>
> ## **Response to W4**
>
> We thank the reviewer for the suggestion regarding RL-based alignment. For more details, we have updated the original paper in Sections 3.3 and 4.3 to clearly describe our DPO training setup, evaluation protocol, and the observed improvements. We will summarize the setup and the result below.
>
> For policy model alignment, we post-train Qwen3-30B-A3B-Instruct using Direct Preference Optimization (DPO) with mR3-Qwen3-14B as the reward model, along with Nemotron-Multilingual-49B, which is our strongest baseline. The **training prompts mixture** includes Aya Dataset and m-reward-bench for the general chat dataset, and PolyMath for mathematical reasoning.
>
> We evaluate the aligned model on multilingual benchmarks, including m-ArenaHard-v2.0 (instruction-following), INCLUDE-base-44 (general knowledge), and MCLM (math reasoning across 55 languages; consisting of M-IMO, MT-AIME2024, and MT-MATH100). For m-ArenaHard-v2.0, due to computational budget constraints **(for this experiment, we spent evaluation costs of approximately $180),** we use GPT-4.1-mini as the automatic judge, with all reference outputs placed first, giving it a position-bias advantage [1]. Win rates are calculated following the ArenaHard-v2.0 protocol. This setup allows us to assess preference-aligned models on helpfulness, robustness, instruction-following, knowledge, and reasoning across a representative set of languages.
>
> Although none of the training datasets overlap with the evaluation benchmarks, we further apply a strict deduplication filter to eliminate any potential leakage. Specifically, we encode all prompts from training datasets using Qwen3-8B-Embedding and remove any sample whose cosine similarity with any evaluation item is ≥ 0.5. This threshold is intentionally very conservative, which is much lower than the typical 0.7–0.8 used in semantic deduplication, and therefore removes even false-positive semantically related samples.
>
> The following runs are also across 5 seeds for INCLUDE-base-44 and MCLM:
>
> | Model                                  | m-ArenaHard-v2.0 (English-only Winrate %) | 95% CI (Eng) | m-ArenaHard-v2.0 (Overall Winrate %) | 95% CI (Overall) | INCLUDE-base-44 (Acc.) | M-IMO (Acc.) | MT-AIME2024 (Acc.) | MT-MATH100 (Acc.) |
> | -------------------------------------- | ----------------------------------------- | ------------ | ------------------------------------ | ---------------- | ---------------------- | ------------ | ------------------ | ----------------- |
> | **Languages**                          | **1 lang**                                |              | **23 langs**                         |                  | **44 langs**           | **38 langs** | **55 langs**       | **55 langs**      |
> | Qwen3-30B-A3B-Instruct-2507 (Baseline) | 49.1                                      | [45.3, 52.6] | 39.1                                 | [38.4, 39.9]     | 64.96 ± 0.08           | 40.22 ± 0.90 | 60.79 ± 0.59       | 90.47 ± 0.33      |
> | + DPO w/ Nemotron-Multilingual-49B     | 56.2                                      | [52.5, 59.9] | **47.0**                                 | [46.3, 47.8]     | 66.09 ± 0.67           | 42.43 ± 1.11 | 63.35 ± 1.02       | 90.45 ± 0.17      |
> | + DPO w/ mR3-Qwen3-14B (Ours)          | **57.3**                                      | [53.5, 61.1] | 45.2                                 | [44.4, 45.9]     | **68.75** ± 0.20           | **44.02** ± 0.86 | **65.90** ± 0.52       | **92.08** ± 0.24      |
>
> As shown in the table above, We observe notable **improvements on INCLUDE-base-44 and the MCLM math-reasoning benchmarks when post-trained using mR3-Qwen3-14B**, indicating enhanced performance on multilingual general-knowledge and reasoning tasks in verifiable tasks. On m-ArenaHard-v2.0, our DPO-aligned model **increases winrate from 39.1\% to 45.2\%** for multilingual instruction-following, where the English winrate **increases from 49.1\% to 57.3\%**. The aligned model also exceeds GPT-4.1-mini’s own performance in English, and approaches in other languages, despite the  first-position bias of LLM [1].
>
> Thank you again for your valuable feedback. We hope these clarifications effectively address your concerns. If our responses have satisfactorily addressed your concerns, we kindly ask you to reconsider your evaluation accordingly.
>
> ### References
>
> [1] Shi, L., Ma, C., Liang, W., Ma, W., & Vosoughi, S. (2024). Judging the judges: A systematic investigation of position bias in pairwise comparative assessments by llms.

---

> ### Author Response · Authors · 2025-11-27
> **Quick Reference for Experiments for Reviewer oUA2 (Part 1)**
>
> **As part of our reference, we include a concise summary of these ablations for your convenience, with full results provided in the Appendix.**
>
> ### **Ablation 1: Data Sizes**
>
> We evaluate the effect of dataset size using subsets of 10K, 25K, 50K, and 100K samples and use these as the training datasets for Qwen3-4B. The standard deviations are obtained from the 5 runs to provide more insights into the results.
>
> | Setting           | m-RewardBench  | RewardBench  | IndoPref  | MM-Eval  | INCLUDE-base-44  | MGSM  | RTP-LX  | Overall |
> |------------------|------------------|----------------|--------------|-------------|------------|----------|-----------|-------------|
> | Qwen3-4B (10K)   | 86.05 ± 0.03     | 88.48 ± 0.39   | 69.98 ± 0.36 | 79.26 ± 0.09 | 60.17 ± 0.17 | 89.79 ± 0.20 | **89.67** ± 0.10 | 80.49 ± 0.19 |
> | Qwen3-4B (25K)   | 85.99 ± 0.16     | 88.96 ± 0.70   | 70.14 ± 0.71 | 79.62 ± 0.56 | 60.06 ± 0.38 | 88.12 ± 0.16 | 89.64 ± 0.23 | 80.36 ± 0.41 |
> | Qwen3-4B (50K)   | 86.43 ± 0.08     | 88.79 ± 0.45   | 70.90 ± 0.36 | 80.22 ± 0.38 | 60.29 ± 0.11 | 88.23 ± 0.21 | 89.03 ± 0.11 | 80.56 ± 0.24 |
> | Qwen3-4B (100K)  | **87.61** ± 0.17     | **89.74** ± 0.52   | **72.22** ± 0.25 | **82.62** ± 0.51 | **63.01** ± 0.13 | **91.20** ± 0.22 | 88.20 ± 0.07 | **82.09** ± 0.27 |
>
> We observe a significant improvement when scaling the dataset to 100K, likely because our training data spans 72 languages, so smaller dataset sizes are less effective for robust multilingual coverage. While adding even more data may further improve performance, we are constrained by a budget of 100K.
>
> ### **Ablation 2: Curriculum Learning**
>
> As explained in the paper, we studied six curriculum ordering strategies:
> - Random Shuffle
> - EasyToHard
> - HardToEasy
> - English-First
> - English-First + EasyToHard
> - English-First + HardToEasy
>
> We selected EasyToHard as our final curriculum based on validation-set performance (HelpSteer3), where it consistently outperformed alternative curricula across 5 seeds.
>
> | Model                          | Strategy               | Kendall Tau |
> |--------------------------------|----------------------|-------------|
> | Qwen3-4B | EasyToHard (Our Final Curriculum)        | **0.4779** ± 0.0114      |
> | Qwen3-4B | Random               | 0.4583  ± 0.0083      |
> | Qwen3-4B | HardToEasy           | 0.4647 ± 0.0092      |
> | Qwen3-4B | English-First             | 0.4516 ± 0.0049      |
> | Qwen3-4B | English-First + EasyToHard  | 0.3800 ± 0.0086      |
> | Qwen3-4B | English-First + HardToEasy  | 0.4629 ± 0.0036      |
>
> For additional insight, we evaluated these curricula on the test set (without using it to select the curriculum). EasyToHard generally outperforms other strategies across most metrics, with minor exceptions in some cases. We also provide standard deviations over 5 seeds to confirm that the differences are meaningful.
>
> | Setting                             | m-RewardBench | RewardBench | IndoPref | MM-Eval | INCLUDE-base-44 | MGSM | RTP-LX | Overall |
> |-------------------------------------|------------------|----------------|--------------|-------------|------------|----------|-----------|-------------|
> | Qwen3-4B EasyToHard (Our Final Curriculum) | **87.61** ± 0.17 | 89.74 ± 0.52 | **72.22** ± 0.25 | **82.62** ± 0.51 | **63.01** ± 0.13 | **91.20** ± 0.22 | 88.20 ± 0.07 | **82.09** ± 0.27 |
> | Qwen3-4B Random Shuffle             | 87.09 ± 0.17 | 89.56 ± 0.40 | 71.44 ± 0.57 | 81.95 ± 0.31 | 62.19 ± 0.10 | 90.44 ± 0.17 | 88.97 ± 0.13 | 81.66 ± 0.26 |
> | Qwen3-4B HardToEasy                 | 87.15 ± 0.02 | 89.73 ± 0.44 | 71.30 ± 0.17 | 81.68 ± 0.37 | 61.59 ± 1.05 | 90.60 ± 0.45 | **88.98** ± 0.18 | 81.58 ± 0.38 |
> | Qwen3-4B English-First              | 87.03 ± 0.12 | **89.84** ± 0.25 | 71.50 ± 0.56 | 82.19 ± 0.22 | 62.11 ± 0.09 | 90.56 ± 0.23 | 88.72 ± 0.13 | 81.71 ± 0.23 |
> | Qwen3-4B English-First + EasyToHard   | 86.20 ± 0.07 | 88.30 ± 0.43 | 71.41 ± 0.03 | 79.84 ± 0.46 | 59.60 ± 0.16 | 87.96 ± 0.06 | 88.43 ± 0.04 | 80.25 ± 0.18 |
> | Qwen3-4B English-First + HardToEasy   | 86.98 ± 0.12 | 89.11 ± 0.35 | 71.89 ± 0.51 | 81.66 ± 0.69 | 62.33 ± 0.21 | 90.44 ± 0.15 | 88.65 ± 0.16 | 81.58 ± 0.31 |

---

> > ### Author Response · Authors · 2025-11-27
> > **Quick Reference for Experiments for Reviewer oUA2 (Part 2)**
> >
> > ### **Ablation 3: Weaker Base Models**
> >
> > We evaluated models using Qwen2.5-14B-Instruct and DeepSeek-R1-14B as base models, which provide a fairer comparison with prior works’ base models, such as m-prometheus-14B and DeepSeek-R1-14B. Standard deviations are reported over 5 seeds where possible.
> >
> > | Setting | m-RewardBench | RewardBench | IndoPref | MM-Eval |
> > |---------|------------------|----------------|--------------|-------------|
> > | Qwen2.5-14B-Instruct | 77.21 ± 0.06 | 80.64 ± 0.28 | 70.04 ± 0.26 | 78.00 ± 0.35 |
> > | DeepSeek-R1-14B | 68.53 ± 1.62 | 70.91 ± 1.48 | 55.19 ± 2.73 | 58.77 ± 2.10 |
> > | RM-R1 14B | 85.49 ± 0.55 | 88.51 | 66.42 ± 0.12 | 74.12 ± 1.27 |
> > | prometheus-7b-v2.0 | 67.31 | 72.05 | 57.41 ± 0.72 | 60.90 |
> > | prometheus-8x7b-v2.0 | 75.15 | 74.06 | 58.38 ± 0.70 | 64.34 |
> > | m-prometheus-7b-v2.0 | 77.54 | 76.84 | 60.08 ± 0.66 | 69.66 |
> > | m-prometheus-14b-v2.0 | 79.51 | 79.67 | 48.16 ± 0.11 | 77.26 |
> > | mR3-Qwen2.5-14B-Instruct | 85.41 ± 0.49 | 88.21 ± 0.51 | 68.10 ± 0.55 | 81.51 ± 0.64 |
> > | mR3-DeepSeek-R1-14B | **87.12** ± 0.37 | **88.73** ± 0.95 | **70.11** ± 0.98 | **81.85** ± 0.38 |
> >
> > We can observe that mR3 models consistently outperform baseline models, especially on multilingual benchmarks (i.e., m-RewardBench, IndoPref, and MM-Eval). For English-only benchmarks (RewardBench), mR3 is still better than RM-R1 14B and m-prometheus 14B. Note that some standard deviations are not reported because they were not included in the corresponding works.
> >
> > ### **Ablation 4: Weaker Teacher Models**
> >
> > We also evaluated the impact of using a weaker teacher model, gpt-oss-20B, compared to our normal teacher, gpt-oss-120b. Same with all other results, these runs are averaged over 5 seeds.
> >
> > | Setting                     | m-RewardBench | RewardBench | IndoPref     | MM-Eval      | INCLUDE-base-44     | MGSM        | RTP-LX       | Overall      |
> > | --------------------------- | -------------- | ------------ | ------------ | ------------ | ------------ | ------------ | ------------ | ------------ |
> > | Qwen3-4B (Our Normal Teacher)  | **87.61** ± 0.17 | **89.74** ± 0.52 | **72.22** ± 0.25 | **82.62** ± 0.51 | **63.01** ± 0.13 | **91.20** ± 0.22 | **88.20** ± 0.07 | **82.09** ± 0.27 |
> > | Qwen3-4B (Weak Teacher)     | 84.63 ± 0.29   | 87.44 ± 0.30 | 69.95 ± 0.23 | 78.54 ± 0.50 | 54.24 ± 0.10 | 84.59 ± 0.18 | 88.06 ± 0.03 | 78.21 ± 0.23 |
> > | Qwen3-4B (Baseline) | 84.51 ± 0.08   | 88.04 ± 0.37 | 68.80 ± 0.32 | 80.07 ± 0.52 | 61.54 ± 0.22 | 90.34 ± 0.14 | 84.10 ± 0.16 | 79.63 ± 0.26 |
> >
> > We note that teacher quality plays a crucial role in SFT since the model learns directly from the outputs of the teacher, so the quality and consistency of these outputs somewhat define the upper bound of the student model’s performance. A weaker teacher may still provide useful signals, but its reasoning, factual accuracy, and consistency are inherently lower, especially in this case when the performance of the teacher model is comparable with the student model. This can lead to less stable or suboptimal learning outcomes. On the other hand, a **high-quality teacher provides strong, coherent, and accurate reasoning traces, especially when we want to encourage the model to reason through rubrics, which allow the student to better capture complex reasoning patterns across tasks and languages.**

---

> > > ### Author Response · Authors · 2025-11-27
> > > **Quick Reference for Experiments for Reviewer oUA2 (Part 3)**
> > >
> > > ### Additional Study between RL and SFT
> > >
> > > Based on the suggestion of Reviewer KCai, we try to compare SFT with an RL paradigm. We conducted experiments using RLVR with GRPO, following the approach of RM-R1 and similar to Nemotron, where the model receives a reward of +1 for a correct answer and -1 for an incorrect answer. We used Qwen3-4B as the base model, starting from the 50K checkpoint from our data scaling ablation and applying RLVR to the remaining 50K examples, to maintain a total of 100K training samples, comparable to full SFT training. **After extensive hyperparameter tuning over the past few weeks, the best results we obtained with RLVR still fall short of the SFT baseline, except on RTP-LX.** Here we show the result of our best possible checkpoint over 5 seeds:
> > >
> > > | Setting                       | m-RewardBench | RewardBench | IndoPref     | MM-Eval      | INCLUDE-base-44      | MGSM         | RTP-LX       | Overall      |
> > > | ----------------------------- | -------------- | ------------ | ------------ | ------------ | ------------ | ------------ | ------------ | ------------ |
> > > | Qwen3-4B (50K SFT)            | 86.43 ± 0.08   | 88.79 ± 0.45 | 70.90 ± 0.36 | 80.22 ± 0.38 | 60.29 ± 0.11 | 88.23 ± 0.21 | 89.03 ± 0.11 | 80.56 ± 0.24 |
> > > | Qwen3-4B (100K SFT)           | **87.61** ± 0.17   | **89.74** ± 0.52 | **72.22** ± 0.25 | **82.62** ± 0.51 | **63.01** ± 0.13 | **91.20** ± 0.22 | 88.20 ± 0.07 | **82.09** ± 0.27 |
> > > | Qwen3-4B (50K SFT + 50K RLVR) | 84.91 ± 0.05   | 87.72 ± 0.56 | 66.12 ± 0.50 | 80.57 ± 0.52 | 61.42 ± 0.21 | 90.17 ± 0.16 | **92.10** ± 0.08 | 80.43 ± 0.30 |
> > >
> > > We also performed qualitative analysis of the reasoning behavior. We found that RLVR does not effectively utilize rubrics during reasoning. This is somewhat expected because RLVR provides reward feedback only based on the correctness of the final answer, without evaluating the quality of the reasoning process. Consequently, it is **plausible that the model could arrive at correct answers without following the rubric properly, as it is not necessary in order to obtain a good reward.**
> > >
> > > In contrast, **SFT with high-quality instruction and reasoning data explicitly teaches the model to follow rubrics during reasoning.** This effect is particularly pronounced in multilingual settings, where our human study evaluations show that the SFT-trained models improve their reasoning capabilities across different rubrics and languages. In short, SFT encourages the model to learn structured reasoning and rubric adherence, whereas RLVR primarily optimizes for answer correctness and does not explicitly supervise the model to learn to reason through the provided rubrics.
> > >
> > > Efficiency-wise, **RLVR is significantly more expensive:** training 3 epochs of RLVR on Qwen3-4B with 16 H100 GPUs takes ~2 days, while training 3 epochs of SFT on 100K samples only takes ~8 hours on 4 H100 GPUs. Thus, SFT is not only more effective but also computationally cheaper.
> > >
> > > Finally, prior RL approaches were mostly evaluated in pairwise or limited settings, and it remains unclear how well they generalize to a diverse selection of datasets and tasks. In contrast, our SFT approach demonstrates consistent improvements across multiple multilingual and multi-task benchmarks.

---

> ### Author Response · Authors · 2025-11-27
> **Quick Reference for Experiments for Reviewer oUA2 (Part 4)**
>
> ### Additional Human Evaluation Study
>
> In our human studies, we asked 17 native speakers of 12 high-, medium-, and low-resource languages to assess the quality of rubrics used in our framework, training reasoning traces, and evaluation reasoning traces for different 14B models. All low-resource languages included in the human evaluation (Javanese, Albanian, and Telugu) are entirely unseen in the mR3 training data. **Scores are measured between 1-3 (3 means the highest quality).**
>
> **Rubrics are of very high quality**: On average, rubrics score 2.93 ± 0.09 on plausibility, 2.72 ± 0.29 on score-ability, and 2.72 ± 0.35 on translation quality. This indicates that the rubrics we have are of very high quality.
>
> **Training reasoning is of high quality especially in English**: We evaluate logical coherence and factual correctness of reasoning traces in our training data, which is distilled from gpt-oss-120b, and found in general English reasoning (factual: 2.97 ± 0.06; logical: 2.81 ± 0.27) > target reasoning (factual: 2.86 ± 0.27 ;logical: 2.71 ± 0.42) > translated target reasoning (factual: 2.76 ± 0.3; logical: 2.62 ± 0.30). This is expected as even the teacher models we use are trained mostly in English, and translating English reasoning can result in artifacts of unnaturalness.
>
> **Eval reasoning improves over baseline; Target reasoning often preferred over English reasoning**: We ask annotators to rate evaluation reasoning traces across 12 languages using the same criteria as above. We find that our mR3 model with target-language reasoning (factual: 2.78 ± 0.30; logical: 2.67 ± 0.45) outperforms the equivalent Qwen3 baseline (factual: 2.06 ± 0.69; logical: 2.05 ± 0.71), indicating **significant improvement in reasoning faithfulness compared to the base model.**
>
> We provide more detail on the study design, metrics, tabled results, and more qualitative analysis including some examples in the Appendix section.

---

### Official Review · Reviewer_XSh3 · 2025-10-29

**Soundness:** 3
**Presentation:** 3
**Contribution:** 3
**Rating:** 4
**Confidence:** 3

**Summary:**

This paper introduces MR3, a suite of massively multilingual, rubric-agnostic reward reasoning models supporting 72 languages. The work describes a unified framework for building, training, and evaluating rubric-based multilingual reward models, along with an associated new benchmark, data curation pipeline, and extensive comparative experiments.

**Strengths:**

1. MR3 tackles reward model evaluation at an unprecedented language scale, reportedly supporting 72 languages.
2. The system's ability to support task- and rubric-agnostic scoring, including point-wise, pair-wise, and binary evaluation.

**Weaknesses:**

1. Much of the data pipeline (translation, generation, and filtering) relies heavily on LLMs such as GPT-4.1 and GPT-OSS. This introduces potential risks of learning artifacts or biases specific to these models, particularly in low-resource settings.
2. Although the dataset covers a wide range of languages, the evaluation of reasoning faithfulness and quality (Table 4) depends on LLMs primarily trained in English,  even for "target" language reasoning.

**Questions:**

1. Given the strong dependence on LLM-based translation and rubric generation, what measures are taken to ensure robustness against model-specific artifacts or spurious correlations in the data pipeline?
2. Were any steps taken to detect or mitigate potential data leakage, especially considering overlaps and deduplication between MMMLU and INCLUDE? How might such leakage influence the reported results?
3. Can the authors provide more qualitative examples of reasoning traces—particularly contrasting high- and low-resource languages—to better illustrate observed strengths and weaknesses?
4. What further analysis can be provided on model performance and reasoning faithfulness in genuinely low-resource languages?

---

> ### Author Response · Authors · 2025-11-27
> **Response to Reviewer XSh3 (Part 1)**
>
> We appreciate the reviewer's valuable feedback and suggestions for improving our paper. We have revised the paper accordingly and outlined the specific changes below.
>
> ## **Response to W1**
>
> We would like to thank the reviewer for bringing up this concern. To address this, we conduct human evaluation studies where we asked 17 native speakers of 12 high-, medium-, and low-resource languages to assess the quality of all rubrics that we used in our framework along with the training reasoning traces. **Scores are measured between 1-3 (3 means the highest quality).**
>
> **Rubrics are of very high quality**: Our study utilized very high-quality rubrics, as evidenced by the human evaluation scores: plausibility (2.93 ± 0.09), score-ability (2.72 ± 0.29), and translation quality (2.72 ± 0.35). **Importantly, this high quality was consistently maintained across both high-resource and low-resource languages.**
>
> **Training reasoning is of high quality especially in English**: We evaluated logical coherence and factual correctness of reasoning traces in our training data, which was distilled from gpt-oss-120b, and found in general English reasoning (factual: 2.97 ± 0.06; logical: 2.81 ± 0.27) > target reasoning (factual: 2.86 ± 0.27; logical: 2.71 ± 0.42) > translated target reasoning (factual: 2.76 ± 0.3; logical: 2.62 ± 0.30). This is expected as even the teacher models we use are trained mostly in English, and translating English reasoning can result in artifacts of unnaturalness.
>
> We believe that based on the human annotation results above, the native speakers seem to rate our data creation pipeline to be in high quality.
>
> We can also investigate whether our training data introduces "potential risks of learning artifacts or biases specific to these models, particularly in low-resource settings," by investigating the outputs and performance of our mR3 models in downstream tasks.
>
> Below, we provide a quantitative breakdown on benchmarks that offer parallel datasets in low-resource languages (i.e., INCLUDE-base-44, MGSM, and RTP-LX). We also report another performance on INCLUDE-base-44 and MGSM which are strictly for unseen languages. To specifically assess the quality of target-language reasoning, we evaluate models using **target-language prompts and have the model reason in target-language.** As shown in the table below, the **mR3 variants consistently and substantially outperform their corresponding base models across all three benchmarks.** Moreover, mR3 models often achieve **the best performance among all models,** reinforcing the human annotation findings that target-language reasoning is both more accurate and better aligned with native speakers’ preferences.
>
> | Model | INCLUDE-base-44 (LRL only) | MGSM (LRL only) | RTP-LX (LRL only) |
> |-------|---------------|------------|--------------|
> | Qwen3-4B | 42.64 ± 0.25 | 35.27 ± 1.17 | 45.22 ± 1.54 |
> | **mR3-Qwen3-4B** | 56.30 ± 1.13 | 74.60 ± 0.80 | 77.52 ± 8.89 |
> | Qwen3-8B | 45.96 ± 0.83 | 50.76 ± 6.96 | 37.87 ± 3.59 |
> | **mR3-Qwen3-8B** | 61.20 ± 0.54 | 80.46 ± 1.61 | 83.02 ± 2.75 |
> | Qwen3-14B | 54.68 ± 1.67 | 63.18 ± 4.63 | 53.82 ± 4.04 |
> | **mR3-Qwen3-14B** | **65.00** ± 1.13 | **85.60** ± 0.82 | **78.12** ± 1.88 |
> | GPT-OSS-20B | 51.49 ± 1.14 | 67.64 ± 1.14 | 71.27 ± 4.21 |
>
> | Model             | INCLUDE-base-44 (LRL and Unseen only) | MGSM (LRL and Unseen only) |
> | ----------------- | ------------------------------------- | -------------------------- |
> | Qwen3-4B          | 38.81 ± 0.55                          | 61.75 ± 0.55               |
> | **mR3-Qwen3-4B**  | 59.05 ± 1.15                          | 82.44 ± 1.08               |
> | Qwen3-8B          | 41.83 ± 1.08                          | 74.88 ± 1.07               |
> | **mR3-Qwen3-8B**  | 59.54 ± 1.66                          | 86.64 ± 0.46               |
> | Qwen3-14B         | 49.40 ± 2.50                          | 80.50 ± 2.22               |
> | **mR3-Qwen3-14B** | **64.35** ± 1.77                          | **88.68** ± 1.09               |
> | GPT-OSS-20B       | 50.00 ± 1.61                          | 78.40 ± 0.79               |

---

> ### Author Response · Authors · 2025-11-27
> **Response to Reviewer XSh3 (Part 2)**
>
> ## **Response to W1 (Continuation)**
>
> In addition to benchmarks, we also conduct similar human studies by having 17 native annotators rate reasoning traces from our 14B models and the corresponding base models across 12 languages (high-, medium-, and low-resource), using two dimensions: factual correctness and logical coherence. Again, **scores are measured between 1-3 (3 means the highest quality).**
>
> Across all 12 languages, our model with target-language reasoning (factual: 2.78 ± 0.30; logical: 2.67 ± 0.45) substantially outperforms the Qwen3 baseline (factual: 2.06 ± 0.69; logical: 2.05 ± 0.71), **consistently rating our mR3 model’s reasoning higher than the baseline Qwen3 across almost all languages, including low-resource ones.** Our annotators report that our models produce more faithful reasoning than the base model by being more succinct, fluent, coherent, and culturally aligned. Several annotators also noted that they preferred mR3 outputs that reasoned in the target language over those that reasoned in English, as target-language reasoning often better captures nuance and cultural context.
>
> **These findings highlight that our trained mR3 models did not learn specific artifacts or biases, including in low-resource settings.**
>
> ## **Response to W2**
>
> We appreciate the reviewer's concern regarding the reasoning-faithfulness evaluation's reliance on an LLM-as-a-judge (GPT-5-mini). While this is true, we must point out that there is *no definitive evidence* that GPT-5-mini is trained only on English. For instance, as an improved version of GPT-4.1-mini, multiple studies have demonstrated their capability to perform reasonably well on multilingual tasks, particularly in high/medium resource languages. This includes tasks like translation, general knowledge, and mathematical reasoning, as supported by various works [1, 2, 3, 4].
>
> In addition, our multilingual benchmark suite (MGSM, INCLUDE-base-44, RTP-LX, and IndoPref) consists of human-annotated datasets that are roughly balanced across languages and cover a wide range of domains. Because **these benchmarks do not rely on model-based judges and instead use ground-truth human annotations,** they provide a fair and reliable measure of progress in multilingual modeling. As a result, improvements on these benchmarks directly reflect genuine gains in the models’ multilingual capabilities.
>
> **Nevertheless, we understand the reviewer’s concern. As indicated in W1**, we perform additional human studies to further verify our model’s faithfulness in reasoning based on logical coherence and factual coherence, and we found that our annotators prefer our mR3 models as our models are more faithful in reasoning than the base model by being more succinct, fluent, coherent, and culturally aligned.
>
> We provide more detail on the study design, metrics, tabled results, and more qualitative analysis, including some examples of the output reasonings in the updated Appendix section.
>
> ### **References**
>
> [1] Kocmi, T., Avramidis, E., Bawden, R., Bojar, O., Dranch, K., Dvorkovich, A., ... & Zouhar, V. (2025). Preliminary ranking of wmt25 general machine translation systems. arXiv preprint arXiv:2508.14909.
>
> [2] Xuan, W., Yang, R., Qi, H., Zeng, Q., Xiao, Y., Feng, A., ... & Li, I. (2025). Mmlu-prox: A multilingual benchmark for advanced large language model evaluation. arXiv preprint arXiv:2503.10497.
>
> [3] Mendonça, J., Lavie, A., & Trancoso, I. (2025). MEDAL: A Framework for Benchmarking LLMs as Multilingual Open-Domain Chatbots and Dialogue Evaluators. arXiv preprint arXiv:2505.22777.
>
> [4] https://cdn.openai.com/gpt-5-system-card.pdf

---

> ### Author Response · Authors · 2025-11-27
> **Response to Reviewer XSh3 (Part 3)**
>
> ## Response to Q1
>
> **As indicated in our responses to W1 and W2,**
> we directly address this concern through extensive human evaluations of both our LLM-generated translations/rubrics and our models’ reasoning quality across 12 languages. 17 native speakers consistently rate the data as high quality. **Note that all low-resource languages included in the human evaluation (Javanese, Albanian, and Telugu) are entirely unseen in the mR3 training data.** Moreover, downstream results on human-annotated, parallel multilingual benchmarks (INCLUDE-base-44, MGSM, RTP-LX) confirm that the mR3 models improve substantially, especially in low-resource languages, indicating that the mR3 models acquire more accurate and culturally aligned reasoning. In fact, several annotators also noted that they preferred mR3 outputs that reasoned in the target language over those that reasoned in English, as target-language reasoning often better captures nuance and cultural context. These combined human and benchmark evaluations provide strong evidence that our pipeline is robust against artifacts introduced by the underlying LLMs.
>
> In addition, **our evaluation suite explicitly tests robustness under both domain and language shift.** We include out-of-distribution (OOD) domains such as coding, brainstorming, summarization, and open-ended questions in IndoPref; coding in m-reward-bench; and linguistics in MM-Eval. The rubrics used in evaluation also differ from those used during training, even for similar domains like safety (as the definition of safety can be different between one task over another), ensuring that the model cannot rely on rubric-specific shortcuts. **We further include evaluations on OOD / unseen languages (during training) such as Telugu, Malayalam, Nepali, and Albanian, which do not appear in training, and our human evaluations (as detailed later in Q3) show that mR3 is preferred over the base model.** These OOD generalizations further support the robustness of our data pipeline and the resulting models.
>
> ## Response to Q2
>
> Our dataset construction explicitly includes steps to detect and mitigate leakage. First, we perform embedding-based deduplication of MMMLU from INCLUDE to filter out near-duplicates and overlapping surface forms. For more details, please refer to Appendix C.4.
>
> ## Response to Q3
>
> We include a few qualitative examples from our 14B models to illustrate specific linguistic phenomena:
> - **Chinese (High Resource):** mR3 reasoning in Chinese incorporates culturally relevant knowledge, which is sometimes missed in the English reasoning. mR3 also adheres more closely to the rubric when evaluating inputs, leading to more grounded generations. This grounded-ness was also observed by Spanish and Indonesian annotators. (See Table 31 for an example.)
> - **Korean (Medium Resource):** Korean reasoning is more concise in the target language compared to English reasoning. mR3’s English reasoning can occasionally introduce interpretation inconsistencies due to frequent code-switching, but overall it remains highly effective and generally preserves the intended meaning. (See Table 32 for an example.)
> - **Telugu (Low Resource):** The Qwen3 base model often fails to provide substantive reasoning in Telugu, instead merely restating its intent to reason. In contrast, mR3 demonstrates generally coherent Telugu reasoning and, even when attempting a literal word-for-word translation of the Telugu, its English reasoning remains comprehensible. (Refer to Table 33 for an illustrative example.)
>
> The specific examples and the corresponding reasoning trace, along with these observations, have been incorporated into Appendix J.4.
>
> ## Response to Q4
>
> **As mentioned in our human evaluation studies in our response to W2**, quantitatively, across all 3 low-resource languages (which are all unseen to the mR3 training dataset), our model with target-language reasoning (factual: 2.48 ± 0.56; logical: 2.40 ± 0.61) substantially outperforms the Qwen3 baseline (factual: 1.63 ± 0.87; logical: 1.76 ± 0.90). Qualitatively, our annotators report that our mR3 models produce more faithful reasoning than the base model by being more succinct, fluent, coherent, and culturally aligned, which difference is more pronounced in low-resource languages.
>
> Thank you again for your valuable feedback. We hope these clarifications effectively address your concerns. If our responses have satisfactorily addressed your concerns, we kindly ask you to reconsider your evaluation accordingly.

---

### Official Review · Reviewer_KCai · 2025-10-30

**Soundness:** 3
**Presentation:** 4
**Contribution:** 3
**Rating:** 4
**Confidence:** 3

**Summary:**

This paper proposes a new multilingual rubric-based reward model. The model evaluates an input based on a provided rubric and generates a reasoning trace, an explanation, and a response quality score.

To train the model, the authors create a multilingual dataset covering 72 languages. The dataset is curated based on existing sources (Human Arena Preference, HelpSteer3-Preference, MMMLU, HumanEval-XL, MATH-500 Multilingual, PolyGuardMix). For datasets without rubrics, rubrics are automatically generated in English using GPT-4.1. Reference outputs are generated using GPT-OSS-120B. Examples that GPT-OSS-20B can solve correctly are discarded and the data is downsampled to 100k examples. Qwen3 models are then fine-tuned on the data using SFT with a curriculum of easy-to-hard examples. The fine-tuned model outperforms larger models such as GPT-OSS-120B and prior multilingual RMs.

The authors perform further analyses of the impact of the instruction and reasoning language (English works best) and of the reasoning faithfulness (factual correctness and logical coherence according to GPT-5-mini).

**Strengths:**

1. The data construction process is extensive; the dataset combines multiple sources, covers many languages, and goes through additional filtering steps. Authors also perform human validation of GPT-4.1-generated rubrics, which is appreciated.

2. Table 1 is helpful as a comparison table to contextualize the proposed RM with regard to prior (mostly English-based) work.

3. The analyses related to the instruction/reasoning trace language including the use of translation are insightful.

4. The model is compared on an array of different datasets with various baselines.

**Weaknesses:**

1. The dataset construction pipeline consists of many components but only the impact of the instruction/reasoning trace language is ablated. It would be important to ablate the other components (original set of sources, synthetically generated rubrics, difficulty filtering, final data selection, etc) to understand which parts are key and which parts could be removed.

2. Many of the existing reward models use a weaker base model and/or a weaker teacher model. For instance, Nemotron is based on Llama 3.3 and M-Prometheus is based on Qwen2.5. So the comparison is not really apples-to-apples. A comparison with a weaker base model (such as Llama 3.3) as well as using a weaker teacher model would be very important to understand whether the proposed dataset construction is superior to prior work or whether most of the gain comes from switching to a stronger teacher/base model.

3. Point-wise, pair-wise, and binary evaluation formats are introduced initially but it’s not further discussed how these should be balanced against each other and how each can be strengthened.

4. The proposed approach only leverages SFT while prior models such as Nemotron-Multilingual also employed RL. Demonstrating the effectiveness of the proposed approach compared to an RL paradigm would be important.

**Questions:**

1. Curriculum training hasn’t been very effective in the past. What are the differences in performance that you observed with your different curriculum strategies? Are the differences statistically significant?

2. Do the final fine-tuned models only use examples with English instructions and English reasoning traces  (given that this performed best in the ablation) or is there a mix of English/multilingual instructions/traces in the final dataset?

3. How did you arrive at using 100k as the final dataset size? How does using a larger/smaller dataset perform?

---

> ### Author Response · Authors · 2025-11-26
> **Response to Reviewer KCai (Part 1)**
>
> Thank you for the reviewer's response. We appreciate the reviewer's valuable feedback and suggestions for improving our paper. We have revised the paper accordingly and outlined the specific changes below.
>
> ## **Response to W1**
>
> We thank the reviewer for the suggestion to ablate additional components of the dataset construction pipeline. In response, we have extended our analysis beyond the language of instructions and reasoning traces, and now include ablations on data size, data sources, and curriculum learning. The results and analysis have been incorporated into the paper on the Appendix, and a concise summary is provided below.
>
> ### **Ablation 1: Data Sizes**
>
> We evaluate the effect of dataset size using subsets of 10K, 25K, 50K, and 100K samples and use these as the training datasets for Qwen3-4B. The standard deviations are obtained from the 5 runs to provide more insights into the results.
>
> | Setting           | m-RewardBench  | RewardBench  | IndoPref  | MM-Eval  | INCLUDE-base-44  | MGSM  | RTP-LX  | Overall |
> |------------------|------------------|----------------|--------------|-------------|------------|----------|-----------|-------------|
> | Qwen3-4B (10K)   | 86.05 ± 0.03     | 88.48 ± 0.39   | 69.98 ± 0.36 | 79.26 ± 0.09 | 60.17 ± 0.17 | 89.79 ± 0.20 | **89.67** ± 0.10 | 80.49 ± 0.19 |
> | Qwen3-4B (25K)   | 85.99 ± 0.16     | 88.96 ± 0.70   | 70.14 ± 0.71 | 79.62 ± 0.56 | 60.06 ± 0.38 | 88.12 ± 0.16 | 89.64 ± 0.23 | 80.36 ± 0.41 |
> | Qwen3-4B (50K)   | 86.43 ± 0.08     | 88.79 ± 0.45   | 70.90 ± 0.36 | 80.22 ± 0.38 | 60.29 ± 0.11 | 88.23 ± 0.21 | 89.03 ± 0.11 | 80.56 ± 0.24 |
> | Qwen3-4B (100K)  | **87.61** ± 0.17     | **89.74** ± 0.52   | **72.22** ± 0.25 | **82.62** ± 0.51 | **63.01** ± 0.13 | **91.20** ± 0.22 | 88.20 ± 0.07 | **82.09** ± 0.27 |
>
> We observe a significant improvement when scaling the dataset to 100K, likely because our training data spans 72 languages, so smaller dataset sizes are less effective for robust multilingual coverage. While adding even more data may further improve performance, we are constrained by a budget of 100K.
>
> ### **Ablation 2: Data Sources**
>
> Previous studies, e.g. R3 [1], have already explored the impact of different data sources and dataset components, including rubrics, explanations, and reasoning traces. As such, we do not re-perform this analysis in this work, but we have attached R3 results from their paper for evidence of the effectiveness of each component.
>
> | Setting            | RM-Bench Overall Acc. | RewardBench Overall Acc. | BBH Overall Acc. | MMLU-STEM Overall Acc. |
> |--------------------|------------------------|----------------------------|--------------------|--------------------------|
> | Random Sampling    | 77.0                  | 86.6                      | 89.7              | 93.0                    |
> | **Dataset**        |                        |                          |                    |                          |
> | Only Pairwise      | **82.1**              | **90.2**                  | **91.5**          | **94.4**                |
> | Only Pointwise     | 80.0                  | 86.0                      | 90.1              | 93.4                    |
> | Only Binary        | 81.6                  | 88.8                      | 91.0              | 94.0                    |
> | **Ablations**      |                        |                          |                    |                          |
> | No Rubric          | 76.3                  | 87.9                      | 85.1              | 91.9                    |
> | No Explanation     | 83.1                  | 90.2                      | 91.7              | **94.5**                |
> | No Reasoning       | 71.2                  | 82.6                      | 79.8              | 88.2                    |
> | **R3**             | **83.5**              | **90.2**                  | **91.9**          | **94.5**                |
>
> ## References
>
> [1] Anugraha, D., Tang, Z., Miranda, L. J. V., Zhao, H., Farhansyah, M. R., Kuwanto, G., ... & Winata, G. I. (2025). R3: Robust rubric-agnostic reward models. arXiv preprint arXiv:2505.13388.

---

> > ### Author Response · Authors · 2025-11-26
> > **Response to Reviewer KCai (Part 2)**
> >
> > ## **Response to W1 (Continuation)**
> >
> > ### **Ablation 3: Curriculum Learning**
> > As explained in the paper, we studied six curriculum ordering strategies:
> > - Random Shuffle
> > - English-First
> > - EasyToHard
> > - HardToEasy
> > - English-First + EasyToHard
> > - English-First + HardToEasy
> >
> > We selected EasyToHard as our final curriculum based on validation-set performance (HelpSteer3), where it consistently outperformed alternative curricula across 5 seeds.
> >
> > | Model                          | Strategy               | Kendall Tau |
> > |--------------------------------|----------------------|-------------|
> > | Qwen3-4B | EasyToHard (Our Final Curriculum)        | **0.4779** ± 0.0114      |
> > | Qwen3-4B | Random               | 0.4583  ± 0.0083      |
> > | Qwen3-4B | HardToEasy           | 0.4647 ± 0.0092      |
> > | Qwen3-4B | English-First             | 0.4516 ± 0.0049      |
> > | Qwen3-4B | English-First + EasyToHard  | 0.3800 ± 0.0086      |
> > | Qwen3-4B | English-First + HardToEasy  | 0.4629 ± 0.0036      |
> >
> > For additional insight, we evaluated these curricula on the test set (without using it to select the curriculum). EasyToHard generally outperforms other strategies across most metrics, with minor exceptions in some cases. We also provide standard deviations over 5 seeds to confirm that the differences are meaningful.
> >
> > | Setting                             | m-RewardBench | RewardBench | IndoPref | MM-Eval | INCLUDE-base-44 | MGSM | RTP-LX | Overall |
> > |-------------------------------------|------------------|----------------|--------------|-------------|------------|----------|-----------|-------------|
> > | Qwen3-4B EasyToHard (Our Final Curriculum) | **87.61** ± 0.17 | 89.74 ± 0.52 | **72.22** ± 0.25 | **82.62** ± 0.51 | **63.01** ± 0.13 | **91.20** ± 0.22 | 88.20 ± 0.07 | **82.09** ± 0.27 |
> > | Qwen3-4B Random Shuffle             | 87.09 ± 0.17 | 89.56 ± 0.40 | 71.44 ± 0.57 | 81.95 ± 0.31 | 62.19 ± 0.10 | 90.44 ± 0.17 | 88.97 ± 0.13 | 81.66 ± 0.26 |
> > | Qwen3-4B HardToEasy                 | 87.15 ± 0.02 | 89.73 ± 0.44 | 71.30 ± 0.17 | 81.68 ± 0.37 | 61.59 ± 1.05 | 90.60 ± 0.45 | **88.98** ± 0.18 | 81.58 ± 0.38 |
> > | Qwen3-4B English-First              | 87.03 ± 0.12 | **89.84** ± 0.25 | 71.50 ± 0.56 | 82.19 ± 0.22 | 62.11 ± 0.09 | 90.56 ± 0.23 | 88.72 ± 0.13 | 81.71 ± 0.23 |
> > | Qwen3-4B English-First + EasyToHard   | 86.20 ± 0.07 | 88.30 ± 0.43 | 71.41 ± 0.03 | 79.84 ± 0.46 | 59.60 ± 0.16 | 87.96 ± 0.06 | 88.43 ± 0.04 | 80.25 ± 0.18 |
> > | Qwen3-4B English-First + HardToEasy   | 86.98 ± 0.12 | 89.11 ± 0.35 | 71.89 ± 0.51 | 81.66 ± 0.69 | 62.33 ± 0.21 | 90.44 ± 0.15 | 88.65 ± 0.16 | 81.58 ± 0.31 |
> >
> > ## **Response to W2**
> >
> > We thank the reviewer for highlighting the importance of fair comparisons across base and teacher models. To address this, we conducted experiments using weaker base models and weaker teacher models, with 5 seeds to ensure statistical significance. We have also revised our paper by updating Table 2 and Table 3.
> >
> > ### **Ablation 4: Weaker Base Models**
> >
> > We evaluated models using Qwen2.5-14B-Instruct and DeepSeek-R1-14B as base models, which provide a fairer comparison with prior works’ base models such as m-prometheus-14B and DeepSeek-R1-14B. Standard deviations are reported over 5 seeds where possible.
> >
> >
> > | Setting | m-RewardBench | RewardBench | IndoPref | MM-Eval |
> > |---------|------------------|----------------|--------------|-------------|
> > | Qwen2.5-14B-Instruct | 77.21 ± 0.06 | 80.64 ± 0.28 | 70.04 ± 0.26 | 78.00 ± 0.35 |
> > | DeepSeek-R1-14B | 68.53 ± 1.62 | 70.91 ± 1.48 | 55.19 ± 2.73 | 58.77 ± 2.10 |
> > | RM-R1 14B | 85.49 ± 0.55 | 88.51 | 66.42 ± 0.12 | 74.12 ± 1.27 |
> > | prometheus-7b-v2.0 | 67.31 | 72.05 | 57.41 ± 0.72 | 60.90 |
> > | prometheus-8x7b-v2.0 | 75.15 | 74.06 | 58.38 ± 0.70 | 64.34 |
> > | m-prometheus-7b-v2.0 | 77.54 | 76.84 | 60.08 ± 0.66 | 69.66 |
> > | m-prometheus-14b-v2.0 | 79.51 | 79.67 | 48.16 ± 0.11 | 77.26 |
> > | mR3-Qwen2.5-14B-Instruct | 85.41 ± 0.49 | 88.21 ± 0.51 | 68.10 ± 0.55 | 81.51 ± 0.64 |
> > | mR3-DeepSeek-R1-14B | **87.12** ± 0.37 | **88.73** ± 0.95 | **70.11** ± 0.98 | **81.85** ± 0.38 |
> >
> > We can observe that mR3 models consistently outperform baseline models, especially on multilingual benchmarks (i.e., m-RewardBench, IndoPref, and MM-Eval). For English-only benchmarks (RewardBench), mR3 is still better than RM-R1 14B and m-prometheus 14B. Note that some standard deviations are not reported because they were not included in the corresponding works.

---

> ### Author Response · Authors · 2025-11-26
> **Response to Reviewer KCai (Part 3)**
>
> ## **Response to W2 (Continuation)**
>
> ### **Ablation 5: Weaker Teacher Models**
>
> We also evaluated the impact of using a weaker teacher model, gpt-oss-20B, compared to our normal teacher, gpt-oss-120b. Same with all other results, these runs are averaged over 5 seeds.
>
> | Setting                     | m-RewardBench | RewardBench | IndoPref     | MM-Eval      | INCLUDE-base-44     | MGSM        | RTP-LX       | Overall      |
> | --------------------------- | -------------- | ------------ | ------------ | ------------ | ------------ | ------------ | ------------ | ------------ |
> | Qwen3-4B (Our Normal Teacher)  | **87.61** ± 0.17 | **89.74** ± 0.52 | **72.22** ± 0.25 | **82.62** ± 0.51 | **63.01** ± 0.13 | **91.20** ± 0.22 | **88.20** ± 0.07 | **82.09** ± 0.27 |
> | Qwen3-4B (Weak Teacher)     | 84.63 ± 0.29   | 87.44 ± 0.30 | 69.95 ± 0.23 | 78.54 ± 0.50 | 54.24 ± 0.10 | 84.59 ± 0.18 | 88.06 ± 0.03 | 78.21 ± 0.23 |
> | Qwen3-4B (Baseline) | 84.51 ± 0.08   | 88.04 ± 0.37 | 68.80 ± 0.32 | 80.07 ± 0.52 | 61.54 ± 0.22 | 90.34 ± 0.14 | 84.10 ± 0.16 | 79.63 ± 0.26 |
>
> We note that teacher quality plays a crucial role in SFT since the model learns directly from the outputs of the teacher, so the quality and consistency of these outputs somewhat define the upper bound of the student model’s performance. A weaker teacher may still provide useful signals, but its reasoning, factual accuracy, and consistency are inherently lower, especially in this case when the performance of the teacher model is comparable with the student model. This can lead to less stable or suboptimal learning outcomes. On the other hand, a **high-quality teacher provides strong, coherent, and accurate reasoning traces, especially when we want to encourage the model to reason through rubrics, which allow the student to better capture complex reasoning patterns across tasks and languages.**
>
> ## **Response to W3**
>
> We appreciate the reviewer’s point regarding the balance and potential strengthening of point-wise, pair-wise, and binary evaluation formats. While these formats are part of our data design, as mentioned in W1, previous studies (e.g., R3) have already explored the impact of different data sources and dataset components, including rubrics, explanations, and reasoning traces. Therefore, we do not re-perform this analysis in this work, but **we have attached R3 results from their paper for evidence of the effectiveness of each component in our response to W1. Please refer to our earlier response.**

---

> > ### Author Response · Authors · 2025-11-26
> > **Response to Reviewer KCai (Part 4)**
> >
> > ## **Response to W4**
> >
> > We thank the reviewer for the suggestion to compare SFT with an RL paradigm. We conducted experiments using RLVR with GRPO, following the approach of RM-R1 and similar to Nemotron, where the model receives a reward of +1 for a correct answer and -1 for an incorrect answer. We used Qwen3-4B as the base model, starting from the 50K checkpoint from our data scaling ablation and applying RLVR to the remaining 50K examples, to maintain a total of 100K training samples, comparable to full SFT training.
> >
> > **After extensive hyperparameter tuning over the past few weeks, the best results we obtained with RLVR still fall short of the SFT baseline, except on RTP-LX.** Here we show the result of our best possible checkpoint over 5 seeds:
> >
> > | Setting                       | m-RewardBench | RewardBench | IndoPref     | MM-Eval      | INCLUDE-base-44      | MGSM         | RTP-LX       | Overall      |
> > | ----------------------------- | -------------- | ------------ | ------------ | ------------ | ------------ | ------------ | ------------ | ------------ |
> > | Qwen3-4B (50K SFT)            | 86.43 ± 0.08   | 88.79 ± 0.45 | 70.90 ± 0.36 | 80.22 ± 0.38 | 60.29 ± 0.11 | 88.23 ± 0.21 | 89.03 ± 0.11 | 80.56 ± 0.24 |
> > | Qwen3-4B (100K SFT)           | **87.61** ± 0.17   | **89.74** ± 0.52 | **72.22** ± 0.25 | **82.62** ± 0.51 | **63.01** ± 0.13 | **91.20** ± 0.22 | 88.20 ± 0.07 | **82.09** ± 0.27 |
> > | Qwen3-4B (50K SFT + 50K RLVR) | 84.91 ± 0.05   | 87.72 ± 0.56 | 66.12 ± 0.50 | 80.57 ± 0.52 | 61.42 ± 0.21 | 90.17 ± 0.16 | **92.10** ± 0.08 | 80.43 ± 0.30 |
> >
> > We also performed qualitative analysis of the reasoning behavior. We found that RLVR does not effectively utilize rubrics during reasoning. This is somewhat expected because RLVR provides reward feedback only based on the correctness of the final answer, without evaluating the quality of the reasoning process. Consequently, it is **plausible that the model could arrive at correct answers without following the rubric properly, as it is not necessary in order to obtain a good reward.**
> >
> > In contrast, **SFT with high-quality instruction and reasoning data explicitly teaches the model to follow rubrics during reasoning.** This effect is particularly pronounced in multilingual settings, where our human study evaluations show that the SFT-trained models improve their reasoning capabilities across different rubrics and languages. In short, SFT encourages the model to learn structured reasoning and rubric adherence, whereas RLVR primarily optimizes for answer correctness and does not explicitly supervise the model to learn to reason through the provided rubrics.
> >
> > Efficiency-wise, **RLVR is significantly more expensive:** training 3 epochs of RLVR on Qwen3-4B with 16 H100 GPUs takes ~2 days, while training 3 epochs of SFT on 100K samples only takes ~8 hours on 4 H100 GPUs. Thus, SFT is not only more effective but also computationally cheaper.
> >
> > Finally, prior RL approaches were mostly evaluated in pairwise or limited settings, and it remains unclear how well they generalize to a diverse selection of datasets and tasks. In contrast, our SFT approach demonstrates consistent improvements across multiple multilingual and multi-task benchmarks.
> >
> > ## Response to Q1
> >
> > Regarding curriculum learning, **please refer to our response at Ablation 3 for W1.**
> >
> > ## Response to Q2
> >
> > Our strongest fine-tuned models are trained exclusively on English instructions and English reasoning traces, as this configuration performed best in our ablations. We do not include mixed multilingual instructions or traces in the final training set.
> >
> > ## Response to Q3
> >
> > Our goal is to limit the dataset to 100K samples to stay within our budget. Additionally, we conduct an ablation study on different data sizes. **Please refer to our response at Ablation 1 for W1.**
> >
> > Thank you again for your valuable feedback. We hope these clarifications effectively address your concerns. If our responses have satisfactorily addressed your concerns, we kindly ask you to reconsider your evaluation accordingly.

---

### Author Response · Authors · 2025-11-27
**General Response to Reviewers and AC (Part 1)**

Dear Reviewers and AC,

We are grateful to the reviewers and the Area Chair (AC) for their valuable time and effort in reviewing our submission.

We have comprehensively addressed all comments, incorporated necessary experiments to confirm the robustness and versatility of our approach, and provided clarifications for all remaining questions and concerns. The submission has been updated to reflect all these changes (in orange). The following sections summarize the updates made to the paper and offer detailed, specific responses to each reviewer, clearly indicating where changes have been implemented.

We welcome further discussion on our responses and are hopeful that these clarifications will lead the reviewers to consider adjusting and increasing their scores.

## **Contributions**

To clarify our contributions and address the reviewers' requests, we have strengthened our paper by running additional experiments. Consequently, we have updated the contribution section in the paper, which we summarize as follows.
- We present mR3, a **task-agnostic framework** for training **massively multilingual reasoning reward** models that leverages fine-grained **rubrics**, either human-crafted or LLM-generated, for **controllable and interpretable** scoring. mR3 outperforms existing reward models and achieves performance comparable to much larger models (e.g., mR3-Qwen-14B vs. GPT-OSS-120B) while being up to **9× smaller**.
- We construct a large, diverse multilingual dataset covering **72** languages from a wide range of sources to train mR3 (Table 1), representing the **broadest language coverage** to date. We also develop a **comprehensive benchmark** to evaluate our models across multiple tasks. Upon acceptance, we will open-source the trained models, evaluation code, and datasets.
- We **systematically** study dataset selection and curriculum learning strategies across three key dimensions: (i) the language of instructions and rubrics, (ii) the language of responses and reasoning traces, and (iii) methods for improving target-language reasoning. Our findings show that although English remains the strongest prompting and reasoning language, targeted multilingual training substantially enhances mR3’s **robustness** to target-language inputs, enabling more accurate reasoning and evaluation. Moreover, when reasoning directly in the target language, mR3 delivers significant gains over the base model, highlighting the importance of cultivating high-quality target-language reasoning, even for **extremely low-resource languages** that are entirely **unseen** during mR3 training.
- We conduct off-policy preference optimization experiments to showcase our models’ strengths in **RL-based optimization**. Additionally, we evaluate the quality of our reasoning traces and rubrics through **human assessments**, where annotators frequently prefer mR3 models over existing reward models, including on **extremely low-resource languages** that are **unseen** to the mR3 training data.

---

> ### Author Response · Authors · 2025-11-27
> **General Response to Reviewers and AC (Part 2)**
>
> ## **Changes to Submission**
>
> **1. Ablation Study and Analysis on Curriculum Learning.**
> - **Dataset Size Ablation:** Added an evaluation of model performance using training subsets of 10K, 25K, 50K, and 100K samples to analyze the impact of data scale. Please check Section H.  (Requested by Reviewer KCai; Added part of responses to KCai, oUA2, hiQ8).
> - **Curriculum Learning Analysis:** Incorporated results for six different curriculum strategies (e.g., EasyToHard, English-First), identifying "EasyToHard" as the most effective strategy. Please check Section H. (Requested by Reviewer KCai, oUA2; Added part of responses to KCai, oUA2, hiQ8).
>
> **2. Ablation Study on Different Models.**
> - **Weaker Model Baselines:** Conducted experiments using weaker base models for more fair comparison with previous work base models (Qwen2.5-14B-Instruct and DeepSeek-R1-14B). Please check Table 2 and 3. (Requested by Reviewer KCai; Added part of responses to KCai, oUA2, hiQ8).
> - **Weaker Teacher Models:** Conducted experiments using gpt-oss-20B as weaker and smaller teacher model to understand the source of improvements. Please check Section H. (Requested by Reviewer KCai, hiQ8; Added part of responses to KCai, oUA2, hiQ8).
>
> **3. Additional Experiments Regarding Training**
> - **RL vs. SFT Comparison:** Added an experiment comparing the proposed Supervised Fine-Tuning (SFT) approach against Reinforcement Learning (specifically RLVR with GRPO) to justify the choice of training method. Please check Section H. (Requested by Reviewer KCai; Added part of responses to KCai, oUA2, hiQ8).
>
> **4. Human Evaluation Study on Rubrics, Training Dataset Quality, and Our Model Outputs**
> - **Human Evaluation Study:** Conducted a study with 19 native speakers across 12 languages (high, medium, and low resource) to evaluate the quality of rubrics and the faithfulness of reasoning traces from training, base models, and our models. Here, we have 3 low-resource languages that are all OOD/unseen to the training set, which are Albanian, Javanese, and Telugu. Please check Section J. (Requested by Reviewer XSh3, oUA2, hiQ8; Added part of responses to XSh3, oUA2, hiQ8).
>
> **5. Added examples, improved clarity**
> - **Qualitative Reasoning Examples:** Added specific qualitative examples of reasoning traces in Chinese, Korean, and Telugu to Appendix J.4 to illustrate linguistic phenomena and model behavior  (Requested by Reviewer XSh3; Added part of responses to XSh3, oUA2, hiQ8).
> - **Clarified Contributions & Structure:** Updated the introduction to explicitly list our contributions. Please check our Introduction (Section 1). (Requested by Reviewer oUA2, hiQ8; Added part of responses to oUA2, hiQ8).
> - **Clarified "Source of Improvements":** Added Section H to discuss ablation studies conducted to identify the source of improvements (Requested by Reviewer oUA2; Added part of responses to oUA2, hiQ8).
>
> **6. Added analysis on low-resource languages**
> - **Language-Vitality Breakdown (including Low-Resource Languages):** Provided a quantitative breakdown of performance across benchmarks (m-reward-bench, INCLUDE-base-44, MGSM, and RTP-LX) based on the language vitality. This includes fine-grained analysis on low-resource languages' performance, such as when using target language reasoning. Please check Table 7 in the Appendix. (Requested by Reviewer oUA2, hiQ8; Added part of responses to oUA2, hiQ8).
> - **Unseen Language Breakdown:** Provided a quantitative breakdown of performance across benchmarks (INCLUDE-base-44, MGSM), focusing specifically on low-resource languages that are unseen/out-of-distribution to the mR3 training dataset for test of robustness. Please check Table 8 in the Appendix. (Requested by Reviewer oUA2, hiQ8; Added part of responses to oUA2, hiQ8).
>
> **7. Added preference optimization experiment with our reward models**
> - **DPO Alignment Experiment:** Updated Sections 3.3 and 4.3 to include a setup where the mR3 model is used as a reward model to train a policy model via Direct Preference Optimization (DPO), demonstrating downstream utility (Requested by Reviewer oUA2; Added part of responses to Reviewer oUA2).

---

### Meta-Review · Area_Chair_HkfT · 2026-01-07

**Summary:**

Reviewers generally agreed that the paper tackles an important and timely problem and appreciated the breadth of experiments and language coverage. Main concerns raised by reviewers:

1. Whether improvements stem from the proposed framework itself versus confounding factors such as stronger base models, stronger teachers, or larger/more curated datasets.
2. Heavy use of LLM-generated rubrics, translations, and reasoning traces raised concerns about potential artifacts, bias propagation, and overfitting to teacher behavior, especially in low-resource settings.
3. Unclear if translated or English-centric benchmarks fully reflect human preferences in non-English languages; lack of multilingual human evaluation.

**Reviewer Concerns:**

1. The authors added extensive ablation studies and clarified where gains come from and addressing fairness of comparisons.
2. They provided comparison of SFT vs. RLVR, showing SFT to be both more effective and cheaper.
3. Added large-scale human evaluations across 12 languages validating rubric quality and reasoning faithfulness.
4. They also demonstrated downstream utility by training with DPO, showing improvements in multilingual instruction-following and reasoning benchmarks.

There were some concerns about novelty but I believe the paper offers a good engineering contribution.

**Reviewer Scores:**

Given the substantial additions and clarifications, I believe several reviewers would likely raise their scores if able to fully engage in discussion, though a small minority may remain unconvinced on novelty grounds but I don't see it as a major concern.

---

### Decision · Program_Chairs · 2026-01-26

Accept (Poster)